# Structural and dynamic insights into agonist recognition and function of the thromboxane A₂ receptor

Pawel Krawinski [1,7], Donna Matzov[2,7], Aoife Ryder[1], Kanhaya Lal[3], Dmitry S. Karlov[3], Georges Chalhoub[4,5], Eamon P. Mulvaney [6], B. Therese Kinsella [6], Peter J. McCormick [4,5], Martin Caffrey [1] ✉, Irina G. Tikhonova [3] ✉ & Moran Shalev-Benami [2] ✉

The thromboxane A₂ receptor (TP), expressed in platelets and smooth muscle, plays an important role in blood clotting and muscle contraction. The endogenous ligand of this G protein-coupled receptor (GPCR), thromboxane A₂ (TXA₂), is a short-lived arachidonic acid metabolite with a half-life of ~30 seconds, which makes investigating the TP structure and activation mechanism highly challenging. Here we determine the structures of the TP in complex with the synthetic agonists, U46619 and I-BOP, stable analogues of the natural ligand, in the presence of the signalling protein partner, Gq. The structures reveal a unique activation switch for the receptor that differs from typical class A GPCR family members. Complemented by functional studies, mutational analysis, docking, and molecular dynamics (MD) simulations, our investigation highlights the differences between agonist and antagonist binding and explores the ligand entry mechanism to the binding pocket from within the membrane via a molecular gate composed of two transmembrane helices. In addition, our study provides crucial information to aid in the rational design of compounds targeting the TP, and offers mechanistic insights into inherited disorders associated with mutations in the TP.

The thromboxane (TX) A₂ receptor (TP) mediates signalling of the prostanoid thromboxane A₂ (TXA₂) and other endogenous ligands to regulate diverse physiological and pathological processes, including platelet aggregation and smooth muscle cell (SMC) contraction[1–4]. The TP is increasingly being recognized to mediate a myriad of other *pro*-inflammatory, *pro*-mitogenic, and *pro*-fibrotic effects in multiple organ systems[3–5]. The receptor is expressed in many cell types, including platelets, SMCs of the vasculature and pulmonary systems, in endothelial cells, cardiomyocytes, monocytes, and macrophages. TP expression is elevated in several pathological conditions, such as cardiovascular and cardiopulmonary diseases[6–13], as well as in certain cancers[14,15]. Accordingly, the TP is an emerging target for clinical development across multiple indications, which include pulmonary arterial hypertension (PAH)[16–19], idiopathic pulmonary fibrosis[20], and related cardiopulmonary conditions[21,22].

[1]School of Medicine and School of Biochemistry and Immunology, Trinity College Dublin, Dublin, Ireland. [2]Department of Chemical and Structural Biology, Weizmann Institute of Science, Rehovot, Israel. [3]School of Pharmacy, Queen's University Belfast, Belfast, UK. [4]Department of Pharmacology and Therapeutics, University of Liverpool, Institute of Systems, Molecular and Integrative Biology, Sherrington Building, Liverpool, UK. [5]Centre for Endocrinology, William Harvey Research Institute, Barts and the London School of Medicine, Queen Mary, University of London, London, UK. [6]ATXA Therapeutics Limited, UCD Conway Institute of Biomolecular and Biomedical Research, University College Dublin, Belfield, Ireland. [7]These authors contributed equally: Pawel Krawinski, Donna Matzov. ✉e-mail: martin.caffrey@tcd.ie; i.tikhonova@qub.ac.uk; moransb@weizmann.ac.il

The TP is a G protein-coupled receptor (GPCR) of the prostanoid receptor subfamily in the class A GPCR family. In humans, two isoforms, referred to as TPα and TPβ, exist, share the same first 328 amino acids and differ in their carboxyl-terminal tails[23–25]. While TPα and TPβ exhibit similar ligand binding and primary heterotrimeric G protein and effector coupling specificities, they have distinct modes of desensitization[12,23]. As their primary signalling pathway[5,23,26–30], TPα and TPβ identically couple with members of the $G_q$ family, where TP stimulation leads to phospholipase (PL)C-β-mediated phosphoinositide hydrolysis, the liberation of inositol 1,4,5-trisphosphate ($IP_3$) and diacylglycerol (DAG), and the subsequent release of intracellular calcium and activation of protein kinase (PK)C, respectively, for downstream signalling[26–29].

Following the determination of the structure of the inactive TP in complex with the antagonists ramatroban and daltroban in 2019[30], the last few years have witnessed the elucidation of a number of high-resolution prostanoid receptor structures. These include active complexes involving $PGE_2$ receptors (EP2, EP3, and EP4 isoforms)[31–33], the $PGF_{2\alpha}$ receptor (FP)[34], as well as the prostacyclin/$PGI_2$ receptor (IP)[35]. Relatedly, while the current study was being finalized, a structure of TPα-$G_q$ in complex with U46619 was published[36]. Collectively, these structures offer insights into the mechanisms of ligand recognition and G protein coupling within the prostanoid receptor subfamily. Despite these advances, the molecular details underlying agonist recognition by the TP and the structural basis of its activation and G protein coupling remain incomplete[36]. In particular, the question of selectivity and potency is not explained. This deficit in knowledge on TP ligand recognition not only hampers the design of highly selective and high-affinity ligands as therapeutic agents, but it also limits our understanding of the physiological and pathological roles of this clinically-important drug target.

In the current study, we report the cryogenic electron microscopy (cryo-EM) structures of the TPα-$G_q$ complex bound to two potent and specific $TXA_2$ mimetics, U46619 and I-BOP. Complemented by functional, mutational, docking, and molecular dynamics (MD) simulations, these structures reveal a mechanism by which agonists selectively engage with and activate the TP, and how outside-in signalling occurs across the membrane through the receptor to its $G_q$ binding partner. In addition, comparisons with antagonist-bound TP structures identify features distinguishing agonist from antagonist binding, and the differences in conformations that influence receptor activation. Finally, our findings also provide mechanistic insights into inherited disorders associated with mutations in the TP.

## Results and discussion

### Cryo-EM structure determination

For structural studies, the full-length human TPα was cloned into a vector encoding an N-terminal haemagglutinin (HA) signal peptide followed by a cleavable FLAG epitope tag. The receptor was coupled to a mini-$G_{\alpha s/q}$ chimera in which the specificity determinants of the α5 helix of $G_{\alpha q}$ were grafted onto mini-$G_{\alpha s}$. A similar construct has been successfully used for the structure determination of the thyrotropin-release hormone receptor 1 (TRH1R)[37], the 5-hydroxytryptamine receptor 2A (H5T2R)[38], and the galanin receptor type 2 (GAL2R)[39]. The first 36 residues of mini-$G_{\alpha q}$ were replaced by the 30 N-terminal amino acids of $G_{\alpha i2}$ to provide a recognition epitope for scFv16 previously shown to stabilize G protein heterotrimer formation[40]. $TXA_2$, the endogenous ligand for the TP, is chemically unstable with a physiological half-life of only 30 seconds[23], which precludes its use in structural determinations and functional studies. The two agonists used in this investigation, U46619 and I-BOP, are both stable $TXA_2$ mimetics, with high specificity and potency at the TP ($EC_{50}$: 11.22 nM and 1.79 nM for U46619 and I-BOP, respectively; Fig. 1). These agonists were introduced to membranes overexpressing the receptor and the $G_q$ heterotrimer. ScFv16 was later added for further complex stabilization. Through cryo-EM, three-dimensional reconstructions were obtained with nominal resolutions of 3.26 Å and 3.25 Å for complexes with U46619 (TP-U46619) and I-BOP (TP-I-BOP), respectively (Fig. 1, Supplementary Figs. 1–2, and Supplementary Table 1). The EM maps revealed the architecture of all complex components including the

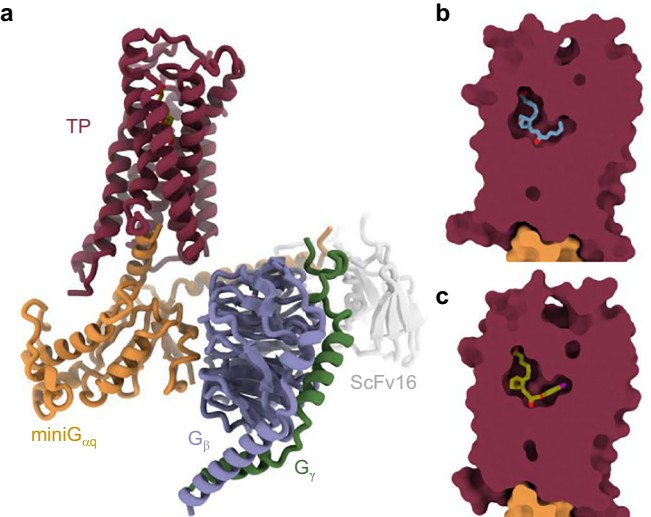

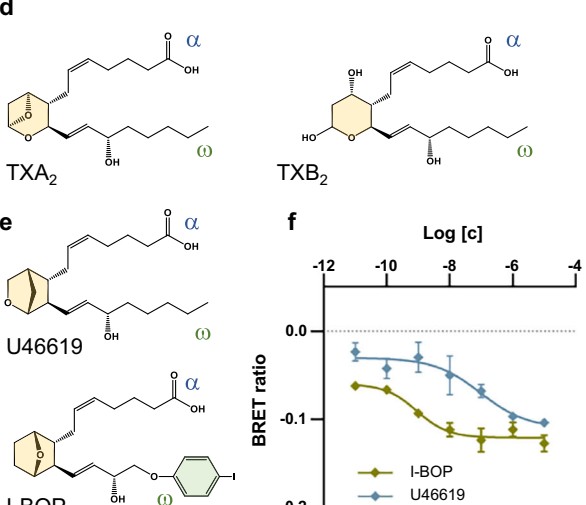

**Fig. 1 | Active state structures and functional characterization of the TP.**
**a** Cartoon representation of the active TP structures bound to the agonist I-BOP and the mini-$G_q$ heterotrimer. The TP is coloured dark red, with mini-$G_{\alpha q}$ in gold, $G_\beta$ and $G_\gamma$ in purple and green, respectively, and scFv16 in light grey. I-BOP is in olive green. Cutaway view of the U46619 (**b**, blue) and I-BOP (**c**, green) bound TP structures indicating that both ligands adopt an L-shaped conformation, while occupying a binding pocket that is buried deep within the receptor's transmembrane core. The sealed cavity is proposed to protect the ligands from decomposition by hydration to the water-soluble $TXB_2$. **d** Chemical structures of the endogenous TP agonist $TXA_2$ (left) and its derivative $TXB_2$ (right). **e** Chemical structures of the synthetic

agonists used in this study: U46619 and I-BOP. In (**d**) and (**e**), the α- and ω-chains are labelled, the bicyclic ring that differs between the synthetic agonists and the endogenous $TXA_2$ is in yellow, and the iodobenzene group present in I-BOP is in green. **f** $G_q$ heterotrimer dissociation as measured by a loss of BRET after TP stimulation with U46619 (blue) or I-BOP (green) in HEK293 cells transiently co-expressing the wild-type receptor and TRUPATH biosensors for the full-length $G_q$ protein. Data are represented as mean ± SEM from 8 independent experiments for U46619 and 4 independent experiments for I-BOP, performed in triplicate, indicating $EC_{50}$ values of 11.22 ± 5.29 nM and 1.79 ± 1.08 nM (mean ± SEM) for U46619 and I-BOP, respectively.

mode of receptor engagement with the $G_q$ heterotrimer and its ligand binding (Fig. 1).

## TP-$G_q$ complex architecture

The structures of the TP-$G_q$ complex bound to I-BOP and U46619 are highly similar, with an all-atom root mean square deviation (RMSD) of 0.5 Å (Supplementary Fig. 2c). In both structures, the receptor adopts an active state conformation, as evidenced by the outward positioning of transmembrane helix 6 (TM6), which allows for engagement with the $G_\alpha$ subunit (Fig. 1a). A notable feature of the TP is extracellular loop 2 (ECL2), consisting of the highly conserved Pro-Gly-Thr-Trp-Cys-Phe-Ile (PGTWCFI) motif, which creates a rigid lid that limits access of ligands to the orthosteric pocket from the extracellular milieu (Fig. 1b, c and Supplementary Figs. 2d, e and 3a). This feature is also found in other prostanoid and lipid GPCRs and is consistent with previous studies suggesting that ligand entry to these receptors occurs predominantly from the membrane[41].

## Ligand-receptor interactions

Like TXA$_2$, U46619 and I-BOP, have a molecular structure consisting of a centrally-located bicyclic ring with monoenoic ($\alpha$-chain) and alkyl chain ($\omega$-chain) modifications (Fig. 1d). The $\alpha$-chain is capped by a carboxyl group, while the $\omega$-chain contains a hydroxyl group and has a terminus that varies from a methyl to more complex ring-containing moieties. Within the ligand-binding pocket, both U46619 and I-BOP assume similar L-shaped conformations, bent at the hydroxyl group, with the extremities of the two chains fixed in position (Figs. 1b, c and 2a).

The carboxylic acid group of the $\alpha$-chain in U46619 and I-BOP forms polar interactions with residues in TM2, TM7, and ECL2. Both agonists are held firmly in place by H89$^{2.65}$ (Ballesteros–Weinstein numbering) and R295$^{7.40}$ at the carboxylate (Fig. 2c, d). In our BRET signalling assays, an R295$^{7.40}$F mutation reduced agonist potency ~200-fold, confirming the importance of the polar interaction between R295$^{7.40}$ and the agonists (Fig. 2b and Supplementary Fig. 4). Constant pH simulations indicated that H89$^{2.65}$ is positively charged in its environment, contributing to additional stabilization of the carboxylate (Supplementary Fig. 5). Accordingly, H89$^{2.65}$Y/F mutations abolished activity, while H89$^{2.65}$R/Q had less impact, suggesting that H89$^{2.65}$ enables key hydrogen bonding (Fig. 2b and Supplementary Fig. 4). For U46619, an additional hydrogen bond is formed between the carboxylate group and S181$^{ECL2}$ (Fig. 2c). The TM7$^{7.40}$ position is a highly conserved arginine in prostanoid receptors, whereas the S181 ECL2-equivalent is either serine or threonine. This conservancy suggests a common role for residues at these positions in interacting with the carboxyl moiety in most prostanoids through polar contacts (Supplementary Fig. 3b). In contrast, the 2.65 position is uniquely a histidine in the TP. The $\alpha$-chain of U46619 and I-BOP is further stabilized by a number of hydrophobic contacts with A31$^{1.39}$, T81$^{2.57}$, V85$^{2.61}$, M112$^{3.32}$, P179$^{ECL2}$, W182$^{ECL2}$, L291$^{7.36}$, L294$^{7.39}$, and T298$^{7.43}$ (Fig. 2c, d). With the exception of A31$^{1.39}$ and T81$^{2.57}$, these positions are mostly conserved in other prostanoid receptors and are in accordance with the similarity of the prostanoid ligands that target them (Supplementary Fig. 3b, c). To evaluate the contribution of hydrophobic interactions to prostanoid activity, we designed two loss-of-function mutations, W182$^{ECL2}$A and L291$^{7.36}$A, and tested the receptor's ability to signal (Fig. 2b and Supplementary Fig. 4). As expected, both mutants were inactive, pointing to the need for large apolar residues at these positions.

The bicyclic rings of U46619 and I-BOP differ from one another in the location of an oxygen atom, which gives rise to an oxane-cyclopentane ring in U46619 and a cyclohexane-oxolane ring in I-BOP (Figs. 1e and 2a). These further differ from the bicyclic oxane-oxetane ring of the endogenous agonist, TXA$_2$ (Fig. 1d). Although no hydrogen bonds are observed between this oxygen and the receptor in either structure, water-mediated interactions occur between the bicyclic ring

oxygen and T298$^{7.43}$ in the course of MD simulations performed for both the endogenous and synthetic agonists (Fig. 2f–h and Supplementary Figs. 6 and 7). Accordingly, in our BRET assays, the T298$^{7.43}$V mutant loses activity, likely due to a lack of hydrogen bond formation (Fig. 2b and Supplementary Fig. 4). Apart from position 7.43, the sub-pocket surrounding the bicyclic ring in the TP is mostly hydrophobic with A31$^{1.39}$, F34$^{1.42}$, C35$^{1.43}$, L78$^{2.54}$, T81$^{2.57}$, G82$^{2.58}$, and V85$^{2.61}$, collectively creating an apolar environment. This characteristic is unique to the TP and is consistent with the lack of hydroxyl or carbonyl groups on the ring in TXA$_2$, which, in contrast to other endogenous prostanoid GPCR agonists, is quite apolar (Supplementary Fig. 3c).

The central ring in prostaglandins is the main feature that distinguishes endogenous prostanoid GPCR agonists from one another, making it an expected determinant of agonist specificity. Accordingly, sequence alignment revealed substantial differences in the residues constituting the ring-binding subpockets of the members of this receptor subfamily (Supplementary Fig. 3c). Collectively, compared to the residues within the TP, in other prostanoid receptors, the corresponding positions are often polar. For example, the residue at position 2.54 is polar in all prostanoid receptors except the TP, where L78$^{2.54}$ is present. Accordingly, in our BRET assay, the L78$^{2.58}$Q mutant showed no activation of $G_q$ signalling, suggesting that the hydrophobicity and/or size of the ring-binding subpocket in the TP is important for agonist binding (Fig. 2b and Supplementary Fig. 4).

The $\omega$-chain of U46619 and I-BOP sits in an apolar subpocket located deep inside the receptor where it forms hydrophobic contacts with L78$^{2.54}$, M112$^{3.32}$, F115$^{3.35}$, G116$^{3.36}$, F184$^{ECL2}$, F200$^{5.43}$, W258$^{6.48}$, L261$^{6.51}$, L294$^{7.39}$, and T298$^{7.43}$. Similar residues were also found in other prostanoid GPCRs (Supplementary Fig. 3d). I-BOP sits deeper in the pocket, especially in the region extending from the ring to the $\omega$-chain hydroxyl group (Fig. 2e). This likely arises, in part, because the $\omega$-chain of U46619, which is composed of methylenes, is more flexible than the rigid $\omega$-chain of I-BOP, which ends with an iodobenzene moiety. The iodobenzene makes closer contacts with F200$^{5.43}$, W258$^{6.48}$, and L261$^{6.51}$. This may contribute to the greater affinity TP has for I-BOP compared with U46619 (Fig. 1e).

Collectively, M112$^{3.32}$ and L294$^{7.39}$ form significant hydrophobic contacts with both the $\alpha$ and $\omega$ chains of the two agonists. By so doing, they may play a role in stabilizing the agonists' L-shaped conformation, which is presumably required for activity. Evidence supporting this proposal was obtained with the M112$^{3.32}$L/L294$^{7.39}$M double mutant and M112$^{3.32}$F and L294$^{7.39}$F single mutants, which showed little activity in our BRET assays (Fig. 2b and Supplementary Fig. 4). The corresponding residues in other prostanoid receptors are also mostly conserved (Supplementary Fig. 3b).

## Comparing agonist and antagonist binding to TP

The X-ray structure of the TP in its inactive form bound to the non-prostanoid antagonists, ramatroban and daltroban, was published in 2019[30]. These antagonists adopt an L-shaped configuration in the inactive receptor similar to that observed for the two agonists in the active TP structures reported in this study (Fig. 3a, b). Notably, the overall length of the agonists exceeds that of the antagonists. The $\alpha$- and $\omega$-chains of the antagonist interact with the binding pocket similarly to agonists. However, additional hydrogen bonds form between the $\omega$-chain sulphonamide in ramatroban and the backbone of L78$^{2.54}$ and the hydroxyl group of T81$^{2.57}$, contributing to placing the antagonist higher in the pocket compared to the agonists (Fig. 3c). These differences in ligand positioning lead to a modified network of interactions with receptor residues that give rise to their differential effects on the receptor. An in-depth description of these interactions and their respective mechanistic implications is provided below.

At the receptor level, the most notable structural rearrangements upon activation include the movement of the extracellular half of TM1 toward TM7, the outward shift of the intracellular half of TM6, and a

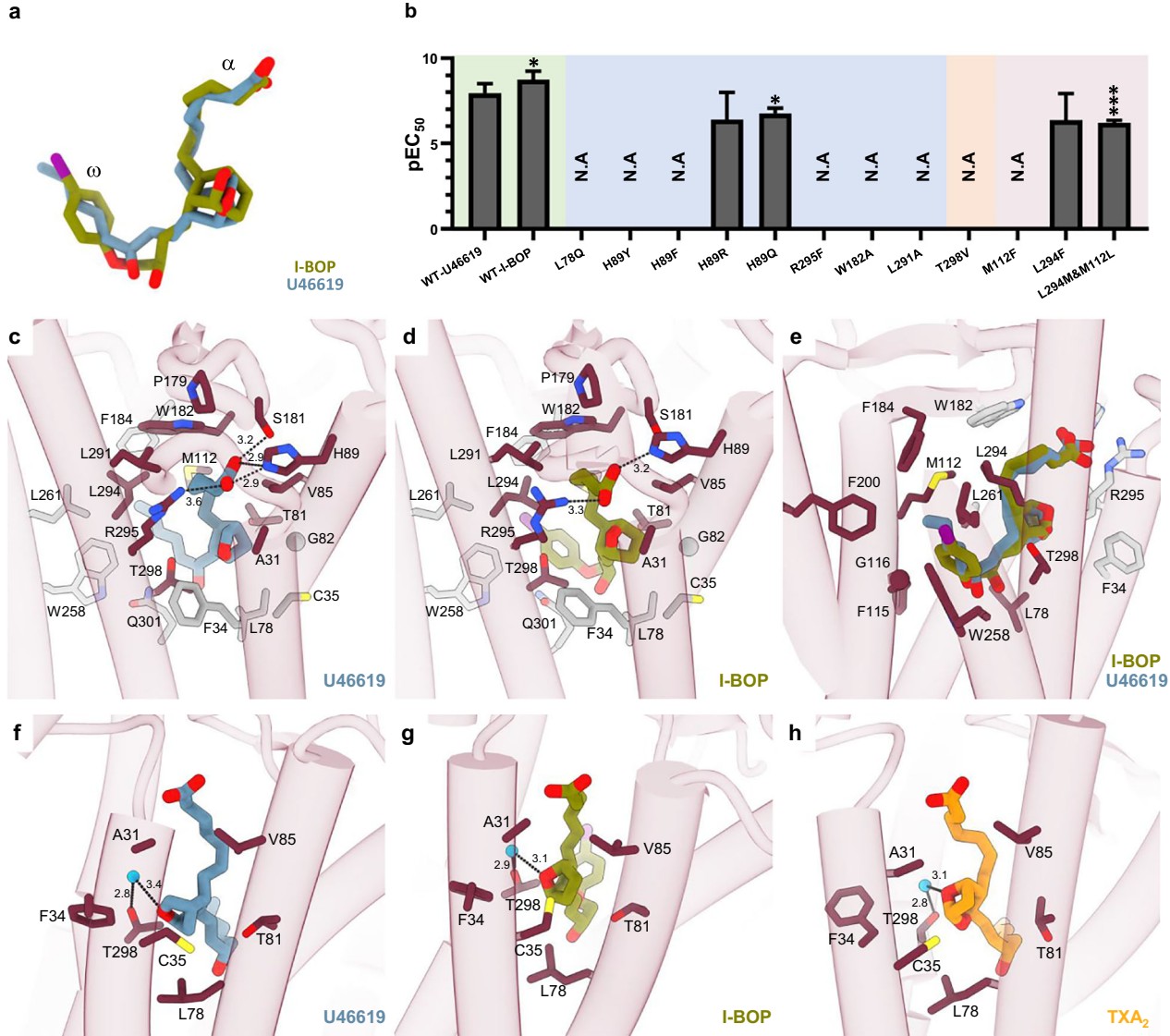

**Fig. 2 | Ligand binding to the TP characterized by cryo-EM, mutational studies, and MD simulations. a** Superposition of U46619 (blue) and I-BOP (green) in the cryo-EM structures. **b** BRET experiments were performed as above and $EC_{50}$ values plotted in response to mutations in the TP canonical binding pocket. $EC_{50}$ values are not available for mutants marked with N.A as their data did not fit dose-response curves. Data are presented as mean ± SEM from three independent experiments. * and *** indicate $P < 0.05$ and $P < 0.001$, respectively. Statistical significance was determined by one-way ANOVA followed by Dunnett's post-hoc tests, comparing the mutants treated with U46619 or I-BOP to the corresponding WT treated with U46619 or I-BOP. Shading indicates the following: WT-TP treated with U46619 and I-BOP (green); mutants of residues involved in agonist binding (blue); the $T298^{7.43}V$ mutant (peach); and mutants of residues involved in stabilizing the L-shaped agonist conformation (salmon). All mutants were treated with U46619. Residues surrounding the binding pocket of U46619 (**c**) and I-BOP (**d**). Residues are labelled and shown as sticks. All residues within 5 Å of the ligand are shown in grey or dark red. Residues that are discussed in the main text are highlighted in dark red. Dashed lines indicate hydrogen bonds with distances in Å. **e** Superposition of U46619 (blue) and I-BOP (green) in the binding pocket of the TP. Only the TP-U46619 receptor structure is shown for clarity. The ω-chain of I-BOP differs from that of U46619 and the natural agonist where the methylenes are substituted by iodobenzene. As a result, I-BOP sits deeper in the binding pocket when compared to U46619. **f–h** Water-mediated interactions, observed through MD simulations, between the bicyclic ring oxygen and $T298^{7.43}$. A more detailed view of the simulation is provided in Supplementary Fig. 6. Simulations are shown for U46619 (**f**, blue), I-BOP (**g**, green), and the endogenous ligand $TXA_2$ (**h**, orange).

rotation of H8 away from TM6 and toward TM1 (Fig. 3d). In addition to these major conformational adjustments, less dramatic structural changes include the outward movement of the cytoplasmic end of TM2 and TM5 and of ICL2, the inward movement of the cytoplasmic end of TM3, and a small shift in the intracellular end of TM1 toward TM2 and away from the receptor core (Supplementary Fig. 8).

### Structural rearrangements that mediate ligand entry
Comparing the structures of the TP in the active and inactive states, a distinctive change at the extracellular face involves TM1 moving towards TM7 (Fig. 3d and Supplementary Fig. 8). In the inactive TP

structures, a gap exists between the extracellular ends of TM1 and TM7. Upon agonist binding, this gap closes. In the active complex, $R295^{7.40}$ on TM7 coordinates with the backbone carbonyl oxygen of $I25^{N-term}$ near the extracellular end of TM1, sealing the gap (Fig. 3d). Similar interactions involving $R295^{7.40}$ have been reported for other prostanoid receptors, such as EP2, EP3 and EP4[32,33,42]. The magnitude of the TM1 movement was found to vary considerably upon activation, with the TP showing the largest movement, followed closely by EP4 (Supplementary Fig. 9a). Notably, the relative movements of TM1 and TM7 have been remarked on for many class A GPCRs that bind hydrophobic ligands, such as the cannabinoid (CB) and sphingosine-1-

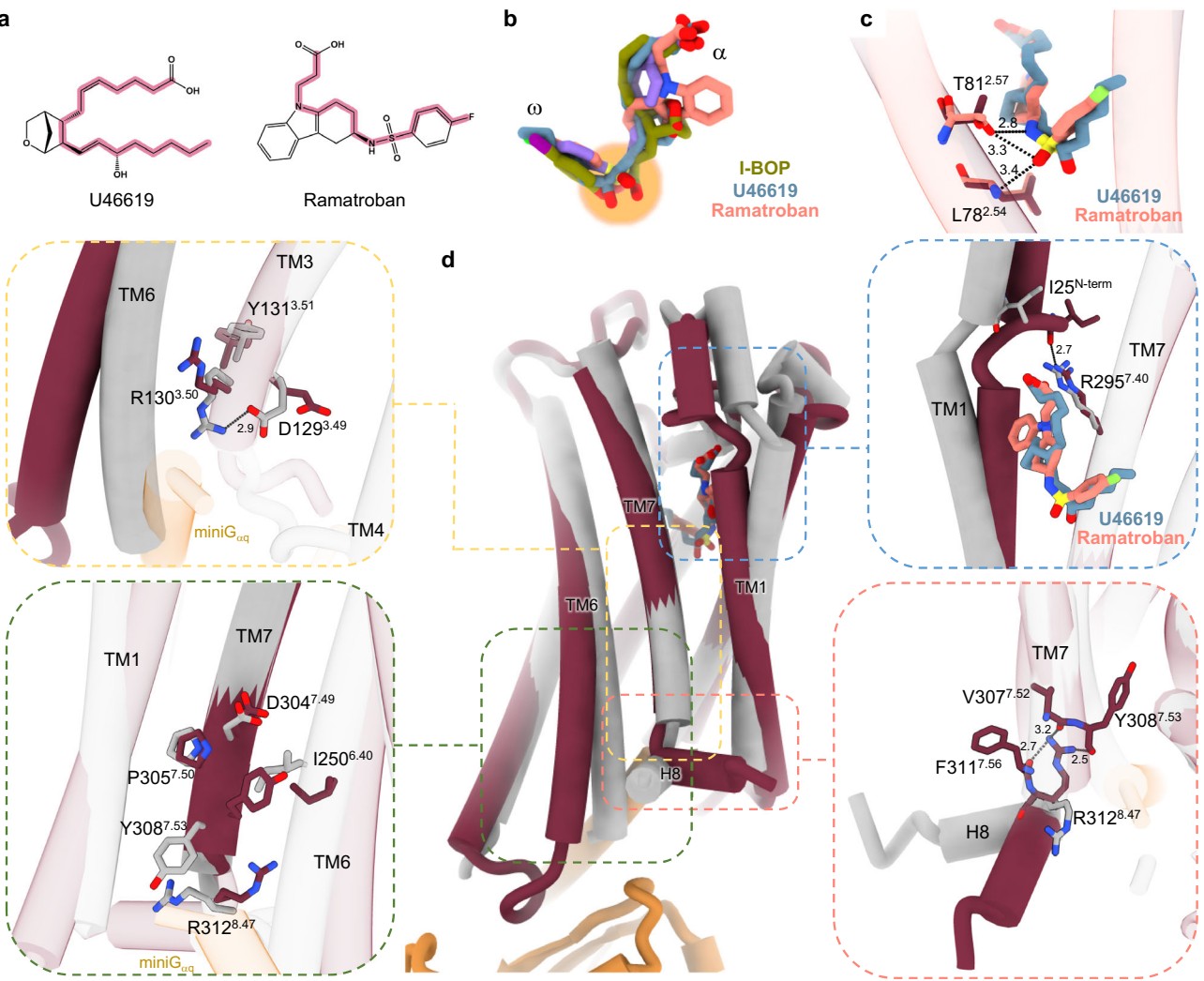

**Fig. 3 | Structural differences in the TP in response to agonist and antagonist binding. a** Chemical structures of the TP agonist, U46619 (left), and antagonist, ramatroban (right). The red outline traces the bonds connecting the carboxyl carbon in the α-chain to the last atom in the ω-chain. There are 16 bonds in the agonist and 14 in the antagonist, indicating that the agonist is the longer ligand. **b** Superposition of the agonists, U46619 (blue) and I-BOP (green), with the antagonist, ramatroban (salmon, PDB 6IIU), indicating that both ligand types adopt an approximate L-shaped conformation in the binding pocket. Agonists sit lower inside the TP binding pocket than antagonists (highlighted by the orange circle).

**c** Superposition of U46619 (blue) and ramatroban (salmon, PDB 6IIU) in their respective binding pockets indicates that the antagonist, ramatroban, contacts the hydroxyl group of T81[2.57] and the backbone of L78[2.54] through the sulphonamide moiety. Hydrogen bonds are indicated as black dotted lines, with distances in Å. **d** Superposition of the active (dark red) and inactive (grey, PDB 6IIU) TP structures highlighting major structural rearrangements in TM1, TM6, TM7, and H8. Close up views of microswitch residues and residues that undergo large conformational rearrangements are shown in dashed outlines.

phosphate receptors[43,44]. This structural rearrangement is considered to play a role in ligand entry, which happens by way of the membrane, rather than via the extracellular milieu. The gap between TM1 and TM7 is considered to serve as a gate that closes behind the ligand upon entry. Overall, the prostanoid GPCR TM1 movements were greater in magnitude than those for other available inactive/active lipid GPCR structure pairs, such as the CB type 1 (CB$_1$) receptor (Supplementary Fig. 9b). The need for enhanced TM1 flexibility and the ability to seal the binding pocket may relate to the labile nature of endogenous prostanoid agonists, which find a stabilizing environment and longevity in the hydrophobic interior of a closed orthosteric pocket.

To gain a greater understanding of the mechanism by which the extracellular end of TM1 undergoes the open/closed transition for ligand entry, we performed functional assays on TP mutants alongside random accelerated MD (RAMD)[45–47] and supervised MD (suMD) simulations[48–50] (Supplementary Figs. 10 and 11). RAMD examines ligand-receptor dissociation in response to a random directional force,

while suMD combines short classical MD simulations with a Tabu-like algorithm, which accepts trajectories when the ligand moves closer to the binding site and rejects them otherwise. Both enhanced sampling methods explore millisecond-scale interactions on nanosecond time-frames. A sequential strategy was employed whereby RAMD was used to study ligand unbinding (computationally efficient but indirect), followed by suMD to examine direct ligand binding, with unbinding results informing ligand placement outside the receptor.

RAMD studies revealed that bulky antagonists (ramatroban and daltroban) primarily dissociated from the receptor via the TM1/TM7 gap, while flexible agonists (U46619 and I-BOP) showed less specific dissociation patterns using both TM1/TM2 and TM1/TM7 openings with considerable frequency and the extracellular route to a lesser extent (Supplementary Fig. 10). These results are consistent with ligand binding to the receptor that is primarily membrane-mediated and are also in agreement with other structures indicating a TM1/TM2 opening that facilitates the entry of lipidic ligands[51]. To avoid bias

toward the open TM1/TM7 gap observed in antagonist-bound states, we simulated I-BOP agonist binding from the membrane using suMD, where the TM1/TM7 gap is closed. Twenty starting structures were selected from 1-μs MD simulations of the agonist- and G protein-free receptor, with I-BOP placed in the membrane extracellular leaflet approximately 45 Å from the binding site. Successful suMD trajectories demonstrated that I-BOP entered the binding site primarily through TM1/TM7 (4 out of 6 simulation runs; Supplementary Fig. 11). Notably, in two independent trajectories, a TM1/TM2 opening was observed, where TM1 moved in an opposite manner to the TM7 gate opening, further highlighting TM1 flexibility and its critical role in membrane-mediated ligand binding (Supplementary Fig. 11).

Interestingly, in both agonist-bound structures, two cholesterol molecules were found docked in a cavity at the extracellular leaflet surface of the receptor created by the closed gap (Supplementary Fig. 12). This cavity is composed mostly of hydrophobic residues in TM1, TM6, and TM7 which include W29$^{1.37}$, F30$^{1.38}$, S33$^{1.41}$, F34$^{1.42}$, V37$^{1.45}$, V253$^{6.43}$, V256$^{6.46}$, C257$^{6.47}$, I292$^{7.37}$, V296$^{7.41}$, W299$^{7.44}$, and L303$^{7.48}$ (Supplementary Fig. 12e). When this gap is open in the inactive receptor, the cavity ceases to exist, and the cholesterol molecules are no longer in evidence at that location. Relatedly, a single cholesterol molecule was reported in each of the inactive structure models. However, these were located in the cytoplasmic membrane leaflet next to the receptor, and they did not overlap with either of the cholesterols in the active TP structures (Supplementary Fig. 12a, b). Two phenylalanines, F30$^{1.38}$ and F34$^{1.42}$, in TM1 undergo major conformational changes upon activation, suggesting that they may function as the hydrophobic gate that closes the TM1/TM7 gap. Superposition with the docked cholesterol molecules further indicates that F30$^{1.38}$ in the inactive state structure clashes with cholesterol localization (Supplementary Fig. 12e, f). Single substitutions of the two phenylalanines to alanine affected the efficacy, but not the potency of receptor activation, while the F34$^{1.42}$Q single mutant had reduced potency (Supplementary Fig. 4). However, the double mutants, F30$^{1.38}$A/F34$^{1.42}$A and F30$^{1.38}$A/F34$^{1.42}$Q were dysfunctional (Supplementary Figs. 4 and 12g). These data provide further experimental evidence supporting the importance of the TM1/TM7 gap in receptor activation, while suggesting that cholesterol may play a role in securing the labile ligand within the pocket.

Interestingly, a W29$^{1.37}$C variant in humans is associated with a rare clotting disorder where uncontrolled bleeding occurs after injury. Specifically, cellular assays have shown reduced surface expression for the mutant receptor, as well as disruptions in ligand binding and receptor activation[52]. W29$^{1.37}$ is a part of the cholesterol binding cavity in the active TP structure (Supplementary Fig. 12e, f). Substitution of the bulky, hydrophobic tryptophan with the smaller, polar cysteine residue may impair receptor association with cholesterol and give rise to membrane localization and signalling defects in individuals harbouring the W29C variant. This proposal is consistent with the important modulatory roles that cholesterol has been shown to play in GPCR function[53,54].

## Rearrangements at the intracellular face and engagement with intracellular partners

The outward movement of the cytoplasmic half of TM6 upon receptor activation is a hallmark of class A GPCRs and is crucial for the opening of the intracellular receptor core for $G_{\alpha q}$ binding and initiation of downstream signalling. Comparing the active and inactive TP structures, TM6 displacement is indeed notable, and is in line with what occurs in other class A GPCRs. In the active TP, this displacement involves the breaking of a conserved ionic lock between E129$^{3.49}$ and R130$^{3.50}$ of the conserved E(D)$^{3.49}$R$^{3.50}$Y$^{3.51}$ motif (Fig. 3d). This triggers a cascade of conformational rearrangements at the intracellular side of the receptor that involves residues in TM5, TM6, TM7, and H8. Specifically, in the inactive structures, residues D238$^{6.28}$, E242$^{6.32}$, and

Q246$^{6.36}$ form hydrogen bonds with R317$^{8.52}$, R313$^{8.48}$, and F311$^{7.56}$, respectively. Additionally, R237$^{6.27}$ hydrogen bonds with Y226$^{5.69}$. Most of these polar interactions are lost in the active TP-U46619 structure, except those between E242$^{6.32}$ and R313$^{8.48}$ and between Y226$^{5.69}$ and R237$^{6.27}$. In the TP-I-BOP structure, Q246$^{6.36}$ hydrogen bonds to the backbone carbonyl oxygen of Y308$^{7.53}$, while all of the other aforementioned interactions are lost (Supplementary Fig. 13).

In addition to the TM6 movement, the conserved N(D)$^{7.49}$P$^{7.50}$xxY$^{7.53}$ motif facilitates the inward movement and slight unravelling of the cytoplasmic end of TM7 upon receptor activation. This repositions Y308$^{7.53}$, allowing its side chain to pack against I250$^{6.40}$ in TM6 and freeing R312$^{8.47}$ in H8 to facilitate a clockwise rotation toward TM1 (Fig. 3d). In the inactive state, Y308$^{7.53}$ sterically limits the rotameric freedom of R312$^{8.47}$. The conformational change of Y$^{7.53}$ upon activation is a characteristic of many class A GPCRs. According to our MD simulations, this change is accompanied by major alterations in the water network at the exposed intracellular surface of these receptors (Supplementary Fig. 6).

One of the most notable changes upon TP activation is the 70° rotation of H8 towards the cytoplasmic end of TM1, which is accompanied by a partial unravelling of TM7 and H8 (Fig. 3d). The inactive TP structures, solved by X-ray crystallography, confirm the mechanistic relevance of these changes as H8 is not involved in crystal contacts. The repositioned H8 forms new hydrophobic interactions with TM1 residues, creating a stabilizing interface. The orientation of H8 is secured by a substantial rotation in the side chain of R312$^{8.47}$, enabling it to hydrogen bond with the backbone carbonyl oxygens of V307$^{7.52}$, Y308$^{7.53}$, and F311$^{7.56}$ in the TP-U46619 structure, fastening the base of H8 to TM7. In the TP-I-BOP structure, R312$^{8.47}$ forms polar interactions only with the carbonyl oxygens of V307$^{7.52}$ and Y308$^{7.53}$ (Fig. 3d). To our knowledge, such a large rotation of H8 has not been seen in other class A GPCRs. A similar rotation was observed when the active and inactive EP4 structures were compared, but H8 participated in crystal contacts in the inactive structure of the receptor[32]. Thus, H8 in the TP may act as a key switch by stabilizing TM6 in the inactive state. However, upon receptor activation, the large rotation of H8 enables greater freedom of movement for TM6, allowing TM6 to acquire an active conformation.

## Agonist mediated mechanism of activation

The canonical mechanism of class A GPCR activation includes TM6 relocation for engagement with the α5 helix of $G_{\alpha}$. In many cases, this structural rearrangement involves the ligand contacting a toggle switch composed of a highly conserved tryptophan in TM6 (W$^{6.48}$) and a hydrophobic residue (M/L/F$^{3.36}$) in TM3. In this mechanism, the ligand directly interacts with position 3.36, leading to a shift in the toggle switch, W$^{6.48}$, ultimately causing TM6 movement and receptor activation. In the TP, the residue at 3.36 is G116$^{3.36}$, which because of its size, does not interact with W258$^{6.48}$ in the same way that bulkier residues interact at this position (Fig. 4a). A comparison of the inactive and active TP structures suggests a unique activation mechanism whereby the ω-chain of both U46619 and I-BOP directly contacts W258$^{6.48}$ causing only a small movement of its indole side chain (Fig. 4e). The mechanism further differs from most class A GPCRs, where toggle switch movement is often accompanied by a rearrangement in the P$^{5.50}$I$^{3.40}$F$^{6.44}$ motif, which is absent in the TP, leading to the outward relocation of TM6 and G protein recruitment.

The presence of glycine at position 3.36 is common to many other prostanoid receptors (Supplementary Fig. 3d; DP1, EP1, EP3, FP, and IP), rhodopsins, and other orphan GPCRs (GPR135, GPR149, and GPR160). Previous studies have shown that replacing G$^{3.36}$ in the EP3 receptor with leucine or tryptophan significantly reduced the potency of agonist binding and increased basal activity[41]. Like TXA₂, PGE₂, the endogenous ligand of EP3, has an extended ω-chain, which supports the claim that this region of the active site is likely to accommodate the

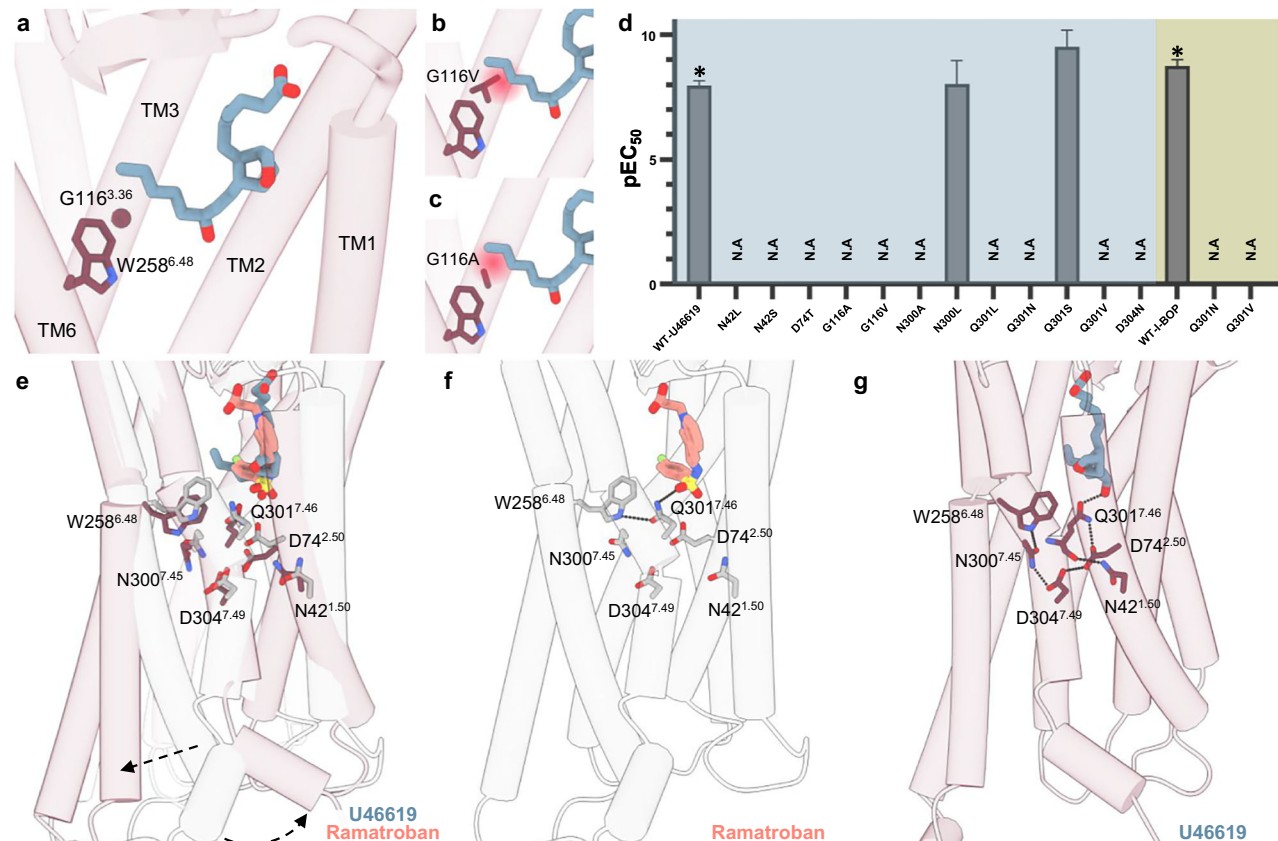

**Fig. 4 | Proposed activation mechanism for the TP. a** U46619 (blue) in the binding pocket of the activated TP, with canonical toggle switch residues highlighted. **b, c** Modelled G116V/A substitutions in the TP sterically clash (red sphere) with the ω-chain of the ligand. **d** EC$_{50}$ values from BRET experiments performed with TP constructs mutated at residues that are part of the proposed activation mechanism. EC$_{50}$ values are not available for mutants marked with N.A as their data did not fit dose-response curves. Data are presented as mean ± SEM from three independent experiments. * indicates $P < 0.05$. Statistical significance was determined by one-way ANOVA followed by Dunnett's post-hoc tests, comparing the mutants treated with U46619 or I-BOP to the corresponding WT treated with U46619 or I-BOP.

Shading indicates the following: WT-TP and mutants of residues involved in the activation mechanism treated with U46619 (blue) and I-BOP (green). **e** Superposition of active (dark red) and inactive (grey, PDB 6IIU) TP structures showing conformational changes in the residues involved in receptor activation. **f** The sulphonamide oxygen of ramatroban hydrogen bonds to the side chain amide of Q301$^{7.46}$, which, in turn, enables the carbonyl oxygen to interact with the indole of W258$^{6.48}$. **g** The ω-chain hydroxyl group of U46619 interacts with the carbonyl oxygen of Q301$^{7.46}$, enabling W258$^{6.48}$ to interact with N300$^{7.45}$ and to form an extensive hydrogen bond network with neighbouring residues that drives receptor activation.

ω-chain. Our BRET assay results with the G116$^{3.36}$A and G116$^{3.36}$V mutants of TP, which were unable to activate G$_q$ (Fig. 4b–d and Supplementary Fig. 4), support this proposal. This suggests an alternative mechanism of activation where W258$^{6.48}$ movement is auxiliary to, or perhaps unnecessary for, TP activation.

Comparing the inactive and active TP, N300$^{7.45}$ and Q301$^{7.46}$ also undergo significant movements (Fig. 4e). In the agonist-bound structure, the agonists sit lower in the pocket, enabling both residues to relocate relative to their positions in the antagonist-bound form. MD simulations reveal that the hydroxyl group in the ω-chain of both agonists forms a hydrogen bond with the carbonyl oxygen of the Q301$^{7.46}$ amide group, while antagonists form a hydrogen bond with the side chain amide of Q301$^{7.46}$ (Fig. 4f, g and Supplementary Fig. 6). This distinct H-bonding pattern enables the carbonyl group of Q301$^{7.46}$ to interact with the indole NH of W258$^{6.48}$ in the antagonist but not in the agonist complexes. The key difference between agonists and antagonists lies in their capacity to form a hydrogen bond as either a donor (agonists) or acceptor (antagonists) with the amide of Q301$^{7.46}$, which, in turn, alters the interaction of Q301$^{7.46}$ with W258$^{6.48}$. The importance of Q301$^{7.46}$ in TP activation is supported by the Q301$^{7.46}$L, Q301$^{7.46}$V, Q301$^{7.46}$S, and Q301$^{7.46}$N mutants, none of which, except Q301$^{7.46}$S, showed G$_q$ activation in BRET assays (Fig. 4d and

Supplementary Fig. 4). Activity of the Q301$^{7.46}$S mutant could be due to residue conservation, as other prostanoid receptors have serine at position 7.46 (Supplementary Fig. 3d).

The agonist-induced repositioning of Q301$^{7.46}$ in the TP is accompanied by the formation of a hydrogen bond between W258$^{6.48}$ and N300$^{7.45}$, as inferred from MD simulations. This interaction is not directly visible in the cryo-EM structures. Nevertheless, these residues are in close proximity, supporting the MD findings. Thus, W258$^{6.48}$ adopts inactive or active conformations by interacting with Q301$^{7.46}$ or N300$^{7.45}$, respectively (Fig. 4f, g). Consistent with an integral role played by N300$^{7.45}$ in activation, the N300$^{7.45}$A mutant was inactive (Fig. 4d and Supplementary Fig. 4). However, N300$^{7.45}$L was active, which could be due to steric hindrance causing W258$^{6.48}$ to move to the active state. This W258$^{6.48}$-Q301$^{7.46}$-N300$^{7.45}$ rearrangement alters the downward interaction network, enhancing interhelical contacts involving TM7. During activation, protonated D74$^{2.50}$ forms a hydrogen bond with D304$^{7.49}$ of the D$^{7.49}$P$^{7.50}$xxY$^{7.53}$ motif, positioning the Q301$^{7.46}$ backbone close to N42$^{1.50}$ thereby enabling hydrogen bond formation (Fig. 4f, g and Supplementary Fig. 6). This draws TM7 closer to TM1, TM2, and TM3, reducing the distance between G123$^{3.43}$ and Y308$^{7.53}$ of the D$^{7.49}$P$^{7.50}$xxY$^{7.53}$ motif indicative of activation (Fig. 4e and Supplementary Fig. 6b). Mutating D74$^{2.50}$ to threonine abolished the agonist

response, confirming its key role (Fig. 4d and Supplementary Fig. 4). Position 2.50 is known to interact with ions and is important for GPCR inactivation, being a highly conserved position across class A GPCRs. Accordingly, established disease-associated mutants N42[1.50]S and D304[7.49]N reduced agonist responses 10- to 200-fold, respectively, supporting the importance of this interaction network in activation (Supplementary Fig. 4). The N42[1.50]S TP variant was identified as the first naturally occurring mutant of the highly-conserved N[1.50] residue in class A GPCRs[55]. A patient with this mutation had significant post-operative and mucocutaneous bleeding, caused by platelet dysfunction due to reduced receptor cell surface expression[55]. A patient with the D304[7.49]N mutation had a history of easy bruising, prolonged epistaxis, and mucocutaneous bleeding, which was attributed to reduced ligand binding to TP[56]. Thus, our structural and mutational data provide a molecular basis for the disorders caused by these mutations.

TM7 side chain rearrangements are coupled to the rotation of H8 upon activation. The movement of TM7 and H8 leads to the untethering of TM6 from these two helices, as hydrogen bonds between Q246[6.36] and the backbone carbonyl oxygen of F311[7.56], as well as those between D238[6.28] and R317[8.52], are broken. The movements of TM7 and H8 are accompanied by an upward rotation of the side chain of R312[8.47] on H8, positioning it to hydrogen bond with the backbone carbonyl oxygens of V307[7.52], Y308[7.53], and F311[7.56] in TP-U46619, or with the carbonyl oxygens of V307[7.52] and Y308[7.53] in TP-I-BOP. These interactions fix H8 angled toward TM1 in the active state. The attendant loss of the D238[6.28]-R317[8.52] and Q246[6.36]-F311[7.56] hydrogen bonds enables the outward movement of TM6. This happens in conjunction with Y308[7.53] interacting hydrophobically with I250[6.40], contributing to TM6's outward migration and a rupturing of the strong ionic lock between R130[3.50] and E240[6.30]. In the TP-I-BOP structure, a further loosening of TM6 from the TM helical core arises from the outward movement of TM5, which breaks the hydrogen bond between Y226[5.69] and the backbone amino and carbonyl groups of R237[6.27]. These combined movements, triggered by agonist binding and relayed predominantly along TM7, open up the receptor's intracellular groove for engagement with the α5 helix of $G_{\alpha q}$ for subsequent signalling.

## Summary

The structures reported herein reveal the active-state TP bound with two TXA$_2$ mimetics, and in complex with its primary G protein signalling partner, providing unprecedented insights into agonist-receptor-effector interactions. Combined with comparisons to existing antagonist-bound TP complexes, MD simulations, and mutagenesis studies, these structures shed light on the molecular bases of disease-associated TP mutations.

Of particular note, the findings of this study identify a unique GPCR activation mechanism for the TP. Rather than involving the highly-conserved W258[6.48] toggle switch seen in many class A GPCRs, Q301[7.46] on TM7 may provide this role in the TP. Antagonists, which serve as hydrogen bond acceptors, do not interact with the carbonyl group of Q301[7.46], which is free to hydrogen bond to W258[6.48] and stabilize an inactive receptor conformation. Coincident stabilizing contacts between TM3, TM5, TM7, and H8 sequester TM6 to the receptor core, sealing the G protein recruitment site. In contrast, agonists act as hydrogen bond donors, sitting deeper in the binding pocket, which enables them to interact with Q301[7.46]. This severs interaction of Q301[7.46] with W258[6.48], facilitating the formation of an extensive hydrogen bond network with residues in TM1, TM2, and TM7. This is accompanied by an unwinding of the intracellular end of TM7, a large rotation of H8, and the breaking of ties securing TM6 to surrounding helices. Consequently, TM6 moves away from the core, opening the α5 binding groove to enable G protein coupling and downstream signalling.

Our structural, mutational, and computational analyses reveal that the TP employs a membrane-mediated ligand entry mechanism facilitated by TM1 flexibility that creates two alternative entry gates for the agonists, namely TM1/TM7 and TM1/TM2. The dynamic TM1/TM7 gating system, regulated by hydrophobic gatekeepers F30[1.38] and F34[1.42], works in concert with cholesterol molecules to accommodate and stabilize labile prostanoid ligands in the binding pocket of the receptor. The clinical relevance of this gating mechanism is highlighted by the bleeding disorder-associated W29[1.37]C variant, where potentially disrupted cholesterol interactions compromise receptor function. While the proposed ligand entry mechanism from MD simulations remains speculative, given the uncertainties about the structure/s of the ligand-free receptor, these insights into the unique structural plasticity of TP provide new opportunities for designing selective therapeutics.

## Methods

### Expression of TP-G$_q$

*Spodoptera frugiperda* (Sf9) insect cells (Oxford Expression Technologies, 600100) were infected with TP and G$_q$ P2 baculoviruses at 100x and 50x viral dilutions, respectively. Cells were cultured in ESF921 medium (without FBS and antibiotics) and infected at a density of 4-5 million cells/mL. Co-expression was carried out for 60 h at 27 °C and 130 rpm. Cells were harvested by centrifugation at 4000 × $g$ for 15 min and then stored at −70 °C.

### Purification of TP-G$_q$ with U46619 and I-BOP

Approximately 70 grams of wet cell mass were lysed at room temperature (20–22 °C) using lysis buffer containing 20 mM HEPES-NaOH (pH 7.5), 100 mM NaCl, 10 mM MgCl$_2$, 10 μM U46619 or 5 μM I-BOP, 25 mU/mL apyrase, and protease inhibitor cocktail (PIC) containing 1 mM phenylmethylsulfonyl fluoride, 160 μg/mL benzamidine, and 2.5 μg/mL leupeptin. Five millilitres of lysis buffer were used for every 1 g of wet cell mass. After 1 h of lysis under gentle agitation, the mixture was centrifuged at 37,000 × $g$ for 30 min at 4 °C. Cell pellets were combined and homogenized at 4 °C in a Dounce homogenizer with solubilization buffer containing 20 mM HEPES-NaOH (pH 7.5), 100 mM NaCl, 10 mM MgCl$_2$, 10 μM U46619 or 5 μM I-BOP, 0.5 %(w/v) laurylmaltose neopentylglycol (LMNG), 0.1 %(w/v) cholesteryl hemisuccinate (CHS), and PIC, using 5 strokes of a loose-fitting pestle followed by 20 strokes of a tight-fitting pestle. Ten millilitres of solubilization buffer were used for every 1 g of membranes. The homogenized membranes were transferred to a beaker, and solubilization was carried out under gentle agitation for 1.5 h at 4 °C. Non-solubilized material was removed by centrifuging the suspension at 37,000 × $g$ for 30 min at 4 °C. Clarified supernatants were transferred to a beaker, and 10 mL of pre-equilibrated *anti*-FLAG beads were added. The beads were pre-equilibrated with 3 column volumes (CVs) of wash buffer containing 20 mM HEPES-NaOH (pH 7.5), 100 mM NaCl, 2 mM MgCl$_2$, 10 μM U46619 or 5 μM I-BOP, 0.075 %(w/v) LMNG, 0.004 %(w/v) CHS, and 0.025 %(w/v) glyco-diosgenin (GDN). Binding of the protein to the *anti*-FLAG beads was carried out under gentle agitation for 3 h at 4 °C. Beads were transferred to a gravity column and were washed with 20 CVs of wash buffer at 4 °C. The beads were incubated in the column for 30 min at 4 °C with 2 CVs of elution buffer containing 20 mM HEPES-NaOH (pH 7.5), 100 mM NaCl, 2 mM MgCl$_2$, 10 μM U46619 or 10 μM I-BOP, 0.075 %(w/v) LMNG, 0.004 %(w/v) CHS, 0.025 %(w/v) GDN, and 0.25 mg/mL FLAG peptide. The protein was collected and concentrated at 4 °C in a 100 kDa molecular weight cut-off (MWCO) concentrator to -0.5 mL. For TP-I-BOP, excess scFv16 at a 1:1.3 molar ratio was added to the protein before concentration and left overnight (-12 h) at 4 °C without agitation to allow for complex formation. The following day, the protein was concentrated as above. The concentrated protein was prepared for size-exclusion chromatography (SEC) by centrifugation at 17,000 × $g$ for 10 min at 4 °C. The sample was then loaded onto and run at 4 °C on a Superose 6 10/300 GL column with running buffer containing 20 mM HEPES-NaOH (pH 7.5), 100 mM

NaCl, 2 mM MgCl$_2$, 10 μM U46619 or 10 μM I-BOP, 0.00075 %(w/v) LMNG, 0.00015 %(w/v) CHS, and 0.00025 %(w/v) GDN. Fractions containing the TP-G$_q$-ligand complex were pooled, concentrated in a 100 kDa MWCO concentrator (~10 mg/mL for TP-U46619 and ~14 mg/mL for TP-I-BOP), and used immediately for cryo-EM grid preparation.

## Expression and purification of scFv16

*Trichoplusia ni* (Tni) Hi5 insect cells (Expression Systems, 94-002 F) were infected with scFv16 P2 baculovirus at a 100x viral dilution. Cells were cultured in ESF921 medium (without FBS and antibiotics) and infected at a density of 4–5 million cells/mL. Expression was carried out for 60 h at 27 °C and 130 rpm. The medium, containing the secreted scFv16, was clarified by centrifugation at 4000 × $g$ for 15 min at 4 °C and used immediately for purification. The pH of the supernatant was adjusted to 8.2 using 1 M Tris-HCl buffer (pH 8.2) and then supplemented with 1 mM NiCl$_2$, 5 mM CaCl$_2$, and PIC. Upon addition of the reagents, a white precipitate formed. The mixture was left to stir at room temperature for 1 h after which the supernatant containing the scFv16 was clarified by centrifugation at 4000 × $g$ for 15 min at 4 °C and then passed through a 0.45 μm filter. Nickel beads, pre-equilibrated with 3 CVs of wash buffer containing 20 mM HEPES-NaOH (pH 7.5), 100 mM NaCl, and 20 mM imidazole-HCl (pH 7.5), were added to the clarified supernatant. Two millilitres of beads were used per litre of medium. Binding of the protein to the nickel beads was carried out under gentle agitation for 3 h at 4 °C. The beads were transferred to a gravity column and washed with 30 CVs of wash buffer at 4 °C. The protein was eluted from the beads with 2 CVs of elution buffer containing 20 mM HEPES-NaOH (pH 7.5), 100 mM NaCl, and 250 mM imidazole-HCl (pH 7.5) at 4 °C. The eluted protein was concentrated at 4 °C in a 3 kDa MWCO concentrator to ~0.5 mL. The concentrated sample was centrifuged at 17,000 × $g$ for 10 min at 4 °C and then loaded onto and run at 4 °C on a Superdex 200 Increase 10/300 GL column with running buffer containing 20 mM HEPES-NaOH (pH 7.5) and 100 mM NaCl. Fractions containing the protein were pooled, concentrated in a 3 kDa MWCO concentrator to ~4 mg/mL, and supplemented with 20 %(v/v) glycerol. The protein was aliquoted, flash frozen in liquid nitrogen, and stored at −70 °C.

## Activity assays

The TRUPATH BRET signalling assays of wild-type TP and its mutants were performed as previously described[57,58]. One microgram of TP plasmid, Rluc8-G$_{\alpha q}$, G$_{\beta 1}$, and GFP2-G$_{\gamma 1}$ were transiently co-transfected into Human Embryonic Kidney (HEK293) cells (ATCC, CRL-1573) in tissue culture dishes using polyethyleneimine. Cells were grown in Dulbecco's Modified Eagle Medium (DMEM) supplemented with 10 % (v/v) FBS and 1 %(v/v) penicillin/streptomycin in a humidified incubator at 37 °C with 5 %(v/v) CO$_2$. Twenty-four hours after transfection, cells from the tissue culture dishes were used to seed poly-D-lysine-treated white 96-well assay plates at 0.6 × 10$^5$ cells per well. The transfection was carried out for another 24 h. One hour prior to performing BRET assays, cells were starved by exchanging their FBS-supplemented DMEM for media without FBS. The media from the plates was aspirated, and coelenterazine 400a solution that was diluted in assay buffer (Hanks' Balanced Salt Solution supplemented with 10 %(w/v) BSA) was added to each well at a final concentration of 5 μM. The plate was placed in the CLARIOStar Plus microplate reader (BMG Labtech) at 30 °C, and the cells allowed to equilibrate with the coelenterazine 400a solution for ~10 min before adding U46619 or I-BOP to the wells at a final concentration range of 10$^{-11}$–10$^{-5}$ M. The BRET signal was read using 410–480 nm excitation and 515–530 nm emission filters.

## Cryo-EM, model building, and refinement

A volume of 3.5 μL of purified protein sample at 10 mg/mL for TP-U46619 or 14 mg/mL for TP-I-BOP was applied to glow-discharged (90 s, 15 mA, PELCO easiGlow, Ted Pella) holey carbon gold grids (Quantifoil®

R1.2/1.3, 300 mesh). The grids were blotted using a Vitrobot Mark IV (FEI) with 3.5 s blotting time at 22 °C in 100 % relative humidity and vitrified in liquid ethane. A total of 8735 and 3212 micrographs were recorded for the TP-U46119 and TP-I-BOP complexes, respectively, on a Titan Krios electron microscope (ThermoFisher Scientific - FEI) operating at 300 kV at a magnification of 105,000-fold and corresponding to a magnified pixel size of 0.8423 Å. The BioQuantum energy filter (Gatan) was operated with an energy slit width of 20 eV. Micrographs were recorded using a K3 direct electron camera (Gatan) with an exposure rate of ~25.5 electrons Å$^{-2}$ s$^{-1}$ and defocus values ranging from −0.8 μm to −2.0 μm. The total exposure time was 1.49 s, and intermediate frames were recorded in 0.033 s intervals, resulting in an accumulated dose of ~38.4 electrons Å$^{-2}$ and a total of 45 frames per micrograph. Automatic data acquisition was done using EPU 3.6 (FEI, ThermoFisher Scientific).

Patch motion correction, CTF estimation, particle picking, 2D classification, ab initio model reconstruction, and heterogeneous refinement were performed in cryoSPARC v4.2.1[59,60]. Homogeneous groups of particles with well-resolved features were refined using the non-uniform refinement algorithm in cryoSPARC. Particle coordinates with their assigned angles were imported into Relion 4.0[61,62] and were subjected to alignment-free 3D classification to enrich for more refined homogeneous particle populations. Selected particles were then transferred back to cryoSPARC and further refined by non-uniform refinement followed by a local refinement focused on the receptor, excluding the micelle and the G proteins, separately. A flowchart describing data processing steps is available in Supplementary Fig. 1. The final map was reconstructed with a total of 216,383 and 242,712 particles for the TP-U46119 and TP-I-BOP complexes, respectively, and has an indicated nominal resolution of 3.26 Å for the TP-U46619 complex and 3.25 Å for the TP-I-BOP complex. The resolution is based on the gold-standard Fourier shell correlation (FSC) using the 0.143 criterion (Supplementary Fig. 2). Angular distribution for particle projections was plotted using cryoSPARC, local resolution was determined using Phenix 1.20.1[63] and was plotted using ChimeraX 1.8[64].

Initial coordinates were generated using the X-ray structure of the inactive TP (PDB code 6IIU[30]). Ligand coordinates and geometry restraints were generated using eLBOW[65]. Models were initially docked into the EM electrostatic potential map using ChimeraX, followed by iterative manual building in Coot 0.9.8.93[66]. The final models were subjected to global refinement and minimization in real space using *phenix.real_space_refine* implemented in Phenix. MolProbity[67] was used to evaluate model geometry. The final refinement parameters are provided in Supplementary Table 1. Structures have been deposited in the Protein Data Bank (PDB) and Electron Microscopy Data Bank (EMDB) databases with accession codes 9GG5/EMD-51324 and 9GGG/EMD-51331 for TP-U46619 and TP-I-BOP, respectively.

## Molecular dynamics simulations

The PDB structures of the TP bound to U46619, I-BOP, ramatroban, and daltroban were prepared using Schrödinger Maestro 2021-3[68]. Missing loops and side chain atoms were added using knowledge-based homology modelling of the Prime module[69,70] with complete amino acid sequences taken from the UniProt database[71]. The added residues were energy minimized using the 3D builder in Maestro. The N- and C-termini of the receptors and the G protein were capped with acetyl and *N*-methyl groups, respectively. The resulting models were validated using the Protein Reports tool in Maestro, and strong steric clashes were removed by geometry minimization. The protonation states of amino acids at pH 7.4 were predicted using the PROPKA 3.1 module in the Protein Preparation workflow of Maestro[72]. Residue H89 was kept protonated in all simulations, while D74 was protonated only in simulations of agonist-bound complexes[73,74] and the empty (without ligand) form of the same complexes.

The input models for MD simulations were prepared using the CHARMM-GUI server[75–83]. Ligand parameter files were created using the

CHARMM-GUI server employing the CHARMM General Force Field to generate topology and parameter files. The receptor was oriented in a membrane using the PPM server within the CHARMM-GUI[84]. The bilayer membrane, with a solvent layer 22.5 Å thick on each side of the membrane, was composed of 9 different lipids including cholesterol (29.5 %); 1-palmitoyl-2-oleoyl-*sn*-glycero-3-phosphocholine, POPC (27.5 %); 1-palmitoyl-2-oleoyl-*sn*-glycero-3-phosphoethanolamine, POPE (15 %); sphingomyelin, PSM (12 %); 1-palmitoyl-2-oleoyl-*sn*-glycero-3-phosphoserine, POPS (11 %); 1-palmitoyl-2-oleoyl-*sn*-glycero-3-phosphoinositol, POPI (2.5 %); 1-palmitoyl-2-lauroyl-*sn*-glycero-3-phosphocholine, PLPC (1.5 %); ceramide (0.5 %); and 1-palmitoyl-2-lauroyl-*sn*-glycero-3-phosphoglycerol, PLPG (0.5 %) as reported in the literature[85]. The system contained 150 mM NaCl as solvent, and the total number of atoms was between 135,000 and 140,000 for receptor-only systems and between 196,000 and 198,000 atoms for receptor-$G_q$ systems.

Minimization, equilibration, and production were carried out using the PMEMD program from the Amber20 package[86] with the CHARMM36m force field[87], and the TIP3P model[88] for membrane lipids, protein, and water, respectively. The non-bonded interaction cut-off was set to 10 Å. Initially, the energy minimization was performed for 10,000 cycles involving 5000 steps of the steepest descent method followed by 5000 steps of the conjugate gradient approach. Restraints were applied to the heavy atoms of the protein and lipids. The systems were first equilibrated in the NVT ensemble for 25 ps with a target temperature at 310 K using the Langevin thermostat[89] and a friction coefficient of 1.0 ps$^{-1}$. This was followed by 15–20 steps of NPT equilibration (with a 1 fs time step) for 10 ns each using the Monte Carlo (MC) barostat[90] to maintain pressure for the membrane systems at 1.0 bar. In the equilibration, the restraint on the heavy atoms was decreased gradually, and in the final step, only the protein backbone was restrained. Production was done using the Langevin thermostat[89] with a friction coefficient of 1.0 ps$^{-1}$ and the MC barostat[90] with a pressure of 1.0 bar and a time step of 2 fs. Each production run was repeated to generate 3 replicas of 1 μs for the empty and ligand-bound receptor systems and 500 ns for the ligand-bound receptor-$G_q$ systems. Simulation snapshots were saved every 10,000 steps.

The results of simulations were analysed using CPPTRAJ from the Amber20 package and MDAnalysis 2.7.0[91,92]. The residue-ligand interaction energy was calculated with the NAMD2 program[93,94] by using the *namdenergy.tcl* script v1.6 for VMD 1.9.3[95]. Calculations were performed using cut-off and switch parameters of 9 Å and 7.5 Å, respectively. Force field parameters were taken from the CHARMM36m parameter files used for the simulations. Energy calculations were performed using 2000 snapshots of each replica.

## Molecular docking

Protein structures were prepared with the protein preparation module, while the structure of the endogenous $TXA_2$ was assessed with the ligand preparation module of the Schrodinger software. I-BOP from the TP-I-BOP complex was selected as the centre of the docking box. Receptor docking grids were generated with the receptor van der Waals radius scaling of 1.0. Docking poses were obtained and evaluated with the Glide program from Maestro[96,97]. The OPLS_2005 force field was used in all calculations.

## Ligand escape

RAMD simulations were performed following best practice, as described[45–47]. A specific branch of GROMACS 2020.5 was used to enable RAMD simulations[98]. Amber topology and equilibrated coordinate files were converted to GROMACS format using ParmEd[99]. For each ligand, 10 independent 5 ns simulations were run with different random seeds. The temperature was set to 310.15 K using the velocity rescaling thermostat. Verlet integration was used for the equations of motion[100]. The neighbour list was updated every 20 steps with a 12 Å cutoff. To avoid an abrupt change to zero at the cutoff distance, van der

Waals potential was multiplied by a switch function between 10 Å and 12 Å to reach zero smoothly at the cutoff distance. Long-range electrostatics were handled with PME[101]. Temperature coupling was done with the Nosé-Hoover thermostat[102] using separate coupling groups for protein/ligand, solvent/ions, and lipids. Pressure coupling employed the Parrinello-Rahman barostat[103]. All bonds with hydrogen atoms were constrained with the LINCS algorithm[104]. The 10 simulation snapshots generated for each ligand were used to initiate 100 RAMD simulations (10 snapshots × 10 seeds) with a random force of 585 kJ mol$^{-1}$ nm$^{-1}$ applied to the ligand centre of mass every 50 steps. Simulations were run until the ligand-reference point separation exceeded 50 Å.

## Agonist binding

Supervised MD (suMD) simulations were performed following the protocol of Pavan et al.[105] using the script 1.2a and ACEMD engine provided[106]. Twenty starting structures were prepared by first selecting four random frames from agonist- and G protein-free receptor simulations, with I-BOP placed in the membrane upper (extracellular) leaflet approximately 45 Å from binding site residues (A32$^{1.40}$, V36$^{1.44}$, F75$^{2.51}$, L79$^{2.55}$, G82$^{2.58}$, V86$^{2.62}$, A90$^{2.66}$, I113$^{3.33}$, G116$^{3.36}$, L117$^{3.37}$, G180$^{ECL2}$, F184$^{ECL2}$, L259$^{6.49}$, I292$^{7.37}$, R295$^{7.40}$, V296$^{7.41}$, W299$^{7.44}$, and I302$^{7.47}$). Each structure was then embedded in a POPC membrane, solvated with water, and equilibrated at physiological NaCl concentration for 5 ns. Five frames were randomly selected from each equilibration, yielding 20 simulation systems.

The suMD algorithm monitored the distance between the ligand and binding site centers of mass (CoM) during simulation. Critical distances defining the association process endpoint were set as: opt_dist = 0.5 Å, bound_dist = 0.75 Å, and meta_dist = 1.0 Å. The algorithm executes 150,000-step simulations, accepting trajectories where the average ligand-to-binding site CoM distance decreases and rejecting those where it increases. Accepted simulations continue from the endpoint, while rejected simulations restart from the previous checkpoint. The process terminates when any critical distance threshold is reached.

## Evaluation of H89 protonation state

Constant pH simulations were performed using the Amber20 program complex following recommendations provided in web-based tutorials (https://ambermd.org/tutorials/advanced/tutorial33/). Protein and ligand structures were extracted from initially equilibrated complexes containing water, different lipids, and ions. New structures were constructed using CHARMM-GUI, with the sole difference being the use of POPC as the only lipid to facilitate implementation of the Lipid21 force field[107]. The FF10 force field for proteins was used as recommended; Gaff2, Lipid21, and the TIP3P model were employed to derive parameters for the ligand, 1-palmitoyl-2-oleoyl-sn-glycero-3-phosphocholine, and water, respectively. The default CHARMM-GUI equilibration procedure was followed, with the only exception being the extension of the production simulation by 5 ns, which was then used for a set of predefined pH values: 2.0, 3.0, 4.0, 4.5, 5.0, 5.5, 6.0, 6.5, 7.0, 7.5, 8.0, 8.5, 9.0, 10.0, and 11.0. Each simulation was repeated 3 times to obtain converged $pK_a$ values. In the case of ligand-bound structures, simulations were repeated 6 times to allow better estimation of protonation state populations at pH 8.0, 8.5, and 9.0. The constant pH simulation protocol includes attempts to change the protonation state for every single protonatable residue (Asp, Glu, and His in this case) every 100 steps of simulation. If the flip of the protonation state was successful, an additional 100 steps were performed before the next flip attempt. To compute the probability of a protonation state flip at a particular pH value, a generalized Born implicit solvation model[108] was used. Solvent and ions were stripped, and the mbondi2 radii set[109] was used in the solvation energy computation. Deprotonated state populations at different pH values were extracted from output files and used to build titration curves. There are two additional approximations used in the simulation. First, the estimation of transition probabilities

between different protonation states was computed assuming the protein is surrounded by water without lipids. Second, the ligand was assumed to be deprotonated at all times, which is a reasonable assumption since it forms a salt bridge with R295[7,40]. In this case, the $pK_a$ of the carboxyl group of the ligand is about 2[110].

## Reporting summary

Further information on research design is available in the Nature Portfolio Reporting Summary linked to this article.

## Data availability

All data needed to evaluate the conclusions in the paper are present in the paper and/or the Supplementary Materials. Structures in the paper have been deposited in the Protein Data Bank (PDB) under accession codes 9GG5 (TP-U46619); and 9GGG (TP-I-BOP). The cryo-EM maps have been deposited in the Electron Microscopy Data Bank (EMDB) under accession codes EMD-51324 (TP-U46619); and EMD-51331 (TP-I-BOP). The following previously reported structures were used for structure analysis: 6IIU (TP-ramatroban); 6IIV (TP-daltroban); 7D7M (EP4-PGE$_2$); 5YWY (EP4-ONO-AE3-208); 6N4B (CB$_1$-MDMB-fubinaca); and 5U09 (CB$_1$-taranabant). The source data for the MD simulation protocols are available on the Zenodo repository [https://doi.org/10.5281/zenodo.17848900]. Source data are provided with this paper.

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

## Acknowledgements

We thank past and present members of the MS&FB group for their assorted contributions to this study. Dr. Nadav Elad is acknowledged for assisting with cryo-EM data collection. The work was supported, in part, by Science Foundation Ireland grants 16/IA/4435 and 22/FFP-A/10278 (M.C.), the European Union's Horizon 2020 research and innovation program under the European Research Council (ERC, grant no. 949364, M.S.-B), the Blavatnik Foundation (M.C. and M.S.-B.), an Irish Research Council Postgraduate Scholarship (GOIPG/2019/3074, P.K), the Zuckerman STEM Leadership Program (M.S.-B.), and the Biotechnology and Biological Sciences Research Council (U.K.) grant BB/R007101/1 (to I.G.T.). This project made use of computational time on Kelvin-2 supported by Engineering and Physical Sciences Research Council (EPSRC) (grant no. EP/T022175/1 and EP/W03204X/1), and ARCHER2 granted via the UK High-End Computing Consortium for Biomolecular Simulation, HECBioSim (https://www.hecbiosim.ac.uk), supported by EPSRC (grant no. EP/R029407/1 and EP/W03204X/1). M.S.-B. holds the Tauro Career Development Chair in Biomedical Research.

## Author contributions

M.C. and M.S.-B devised the original research plan; P.K produced and purified proteins with the assistance of A.R. and P.K. ran BRET assays with the help of G.C. and under the supervision of P.J.M.; P.K. prepared samples for EM with the help of D.M., and under the supervision of M.S.-B.; D.M. prepared grids, obtained cryo-EM data, processed data, and modelled the structure under the supervision of M.S.-B.; MD simulations and docking were performed by K.L. and D.S.K. under the supervision of I.G.T. M.C., I.G.T., P.J.M., and M.S.-B. supervised research; P.K. wrote the original manuscript with the help of M.C. Revised manuscript versions were prepared by P.K., M.C., I.G.T., E.M., B.T.K., and M.S.-B. All authors provided input for the final version of the manuscript.

## Competing interests

The authors declare no competing interests.
