## [Transparent Peer Review file · Nature Communications]

Structural and Dynamic Insights into Agonist Recognition by and Function of the Thromboxane A₂ Receptor: Implications for Disease-Causing Mutations

Corresponding Author: Dr Moran Shalev-Benami

Version 0:

Reviewer comments:

Reviewer #1

(Remarks to the Author)

- What are the noteworthy results?

We can't say the cryo-EM structure as this was published 6 months ago with the same ligand (U46619) and G protein: Li et al., Cell Rep 54 113893 (2024).

The analysis of the mechanism is comprehensive, far more so than Li et al. (2024).

The unbinding mode of the ligand is significant, as it shows exit into the membrane.

- Will the work be of significance to the field and related fields? How does it compare to the established literature? If the work is not original, please provide relevant references.

Unfortunately, the Cryo EM structure is already published with the same ligand (U46619) and G-protein, so this part may not be significant. (Li et al., Cell Rep 54 113893 (2024), though they did include an extra ligand. In defence of still sending this article to a high quality journal, they claim 'the question of selectivity and potency are not explained'. It is certainly true that the current article offers a far more comprehensive analysis of the mechanism, which is certainly informative.

- Does the work support the conclusions and claims, or is additional evidence needed?

Only simulations of ligand exit have been performed. To justify ligand binding from the membrane, as claimed in the abstract and suggested 5 years ago, the authors should simulate the binding mechanism from the membrane into the receptor, presumably using an enhanced and adaptive sampling method.

- Are there any flaws in the data analysis, interpretation and conclusions? Do these prohibit publication or require revision?

None apart from the discussion of the W6.48 'toggle switch'. The name 'toggle switch' predates active structures of class A GPCRs which failed to show the conformational change implied by the name, and indeed here, we see a displacement but not a conformational change, so the extensive use of the phrase 'toggle switch' is misleading.

- Is the methodology sound? Does the work meet the expected standards in your field?

yes

- Is there enough detail provided in the methods for the work to be reproduced?

yes

Reviewer #2

(Remarks to the Author)

In this manuscript, Pawel Krawinski and colleagues present two Cryo-EM structures of Gq-coupled TP bound to two analogues of the endogenous ligand, U46619 and I-BOP respectively. The structures suggest Q3017.46 on TM7 may act as the toggle switch in the TP, rather than the highly-conserved residue W2586.48 seen in many class A GPCRs, proposing a unique activation mechanism for the receptor. Therefore, these structures could broaden our understanding for activation of class A GPCRs.

However, I have some concerns about the manuscript.

1. the dose-response curves of BRET assay in Fig 1f and some in supplementary Fig 4 are not entirely convincing with big error bars and each panel of mutant curves also need a WT curve as a control for well evaluating the key roles of the

residues involved.

2. the densities of the agonists should be showed alone for evaluating whether they are fitted properly.

3. although the part “Structural rearrangements that mediate ligand entry” is novel and interesting, the data in this manuscript is not enough to support the hypothesis and a short discussion in this part is needed to clarify that that is just one possibility rather than a final conclusion about the ligand entry. Otherwise, stronger evidence should be provided, because the inactive state can't represent the real apo state (no bound to agonist and G protein), in other word, we don't know whether there is an open gate for ligand entry between TM1 and TM7 in its real apo state conformation similar to that of its inactive state.

4. The last but also the most important, the density map and structure models need to be further improved before it can be published in Nature Communications or other journals. Actually, the densities of Helix 8, ECL and side chains are really poor from the supplementary Fig1., which need more polish, otherwise, they can't support the model properly.

Version 1:

Reviewer comments:

Reviewer #1

(Remarks to the Author)

I've already answered these questions

Reviewer #2

(Remarks to the Author)

The revised manuscript has addressed all my questions, and I have no further inquiries.

Version 2:

Reviewer comments:

Reviewer #1

(Remarks to the Author)

Review of resubmission

P7, line 165. Demonstrating is too strong because in L78Q, the Q is bigger than L

P9, 'while SuMD guides the ligand towards the binding site' is a misleading explanation of SuMD that under sells the method (and hence this article) – perhaps the authors should read more references rather than just the original 2014 article.

p 11, I am not convinced that W29 undergoes a major conformational transition – as I see it, it is more that TM1 moves.

Reviewer #3

(Remarks to the Author)

The authors have incorporated an extensive set of molecular dynamics (MD) simulations in the revised manuscript to substantiate the claims arising from the structural and biochemical data. This addition significantly strengthens the mechanistic framework of the study. The computational component is clearly described, methodologically sound, and executed with the required scientific rigor. The combination of conventional MD, random accelerated MD (RAMD), and supervised MD (suMD) simulations is appropriate and well-justified. The analyses were internally consistent and aligned with the experimental findings, providing coherent mechanistic insights into ligand binding and activation of the thromboxane receptor. This study is of interest to researchers in the fields of structural biology, computational biophysics, and GPCR pharmacology. To further enhance transparency and reproducibility, it is recommend that the authors deposit representative MD trajectories, input parameter files, and analysis scripts in a public repository (e.g., Zenodo, Figshare, or GPCRmd) and include accession details in the Data Availability section. This will ensure that the computational work can be independently validated and reused in accordance with current community standards for data sharing in structural and computational biology.

Reviewer #1

• *What are the noteworthy results?* We can't say the cryo-EM structure as this was published 6 months ago with the same ligand (U46619) and G protein: Li et al., Cell Rep 54 113893 (2024). The analysis of the mechanism is comprehensive, far more so than Li et al. (2024). The unbinding mode of the ligand is significant, as it shows exit into the membrane.

We very much appreciate the Reviewer's favorable comments on the analysis of TP's signaling and ligand unbinding mechanisms reported in this study.

• *Will the work be of significance to the field and related fields? How does it compare to the established literature? If the work is not original, please provide relevant references.* Unfortunately, the Cryo-EM structure is already published with the same ligand (U46619) and G-protein, so this part may not be significant. (Li et al., Cell Rep 54 113893 (2024), though they did include an extra ligand. In defense of still sending this article to a high-quality journal, they claim 'the question of selectivity and potency are not explained'. It is certainly true that the current article offers a far more comprehensive analysis of the mechanism, which is certainly informative.

We thank the Reviewer for their favorable comments. in particular, highlighting our work on mechanism.

• *Does the work support the conclusions and claims, or is additional evidence needed?* Only simulations of ligand exit have been performed. To justify ligand binding from the membrane, as claimed in the abstract and suggested 5 years ago, the authors should simulation the binding mechanism from the membrane into the receptor, presumably using an enhanced and adaptive sampling method.

We appreciate the Reviewer's suggestion regarding simulating the binding mechanism from the membrane. While this appears conceptually straightforward, our initial attempts revealed significant technical challenges specific to our system. The agonist-bound complexes present a particularly complex case due to their more constrained binding site compared to antagonist-bound structures. Our preliminary funnel metadynamics simulations with the simpler antagonist system (ramatroban) already required one month of computing time on a dual-GPU workstation and still faced convergence issues. The situation becomes substantially more challenging for agonists, as simulating their binding would require simultaneous sampling of both ligand movements and protein conformational changes, particularly the TM1/TM7 dynamics. The choice of collective variables for enhanced sampling methods poses additional complications - they would inherently bias the pathway selection and might force the ligand through available gaps simply because they exist in the starting structure, rather than revealing the true binding pathway. Given these technical limitations, we believe our current RAMD approach, while not perfect, provides valuable insights into possible ligand entry/exit pathways without introducing artificial biases from enhanced sampling methods. We agree that direct simulation of binding would be ideal, but current computational methods and resources make this unfeasible for our specific system without compromising the reliability of the results.

All of the Reviewers' comments and Author Responses will be posted and available to the Reader in the Peer Review File upon publication.

• *Are there any flaws in the data analysis, interpretation and conclusions? Do these prohibit publication or require revision?* Nonapparent, apart from the discussion of the W6.48 'toggle

switch'. The name 'toggle switch' predates active structures of class A GPCRs which failed to show the conformational change implied by the name, and indeed here, we see a displacement but not a conformational change, so the extensive use of the phrase 'toggle switch' is misleading.

We agree with the Reviewer that use of the term 'toggle switch' can be misleading. Accordingly, the Main and Supplementary texts have been revised. Reference to 'toggle switch' is only made when discussing receptors where the descriptor is appropriate.

• *Is the methodology sound? Does the work meet the expected standards in your field?*
yes

• *Is there enough detail provided in the methods for the work to be reproduced?*
yes

Reviewer #2

In this manuscript, Pawel Krawinski and colleagues present two Cryo-EM structures of Gq-coupled TP bound to two analogues of the endogenous ligand, U46619 and I-BOP respectively. The structures suggest Q301^{7,46} on TM7 may act as the toggle switch in the TP, rather than the highly-conserved residue W258^{6,48} seen in many class A GPCRs, proposing a unique activation mechanism for the receptor. Therefore, these structures could broaden our understanding for activation of class A GPCRs.

We thank the Reviewer for their favorable comments.

However, I have some concerns about the manuscript.

1. the dose-response curves of BRET assay in Fig 1f and some in supplementary Fig 4 are not entirely convincing with big error bars and each panel of mutant curves also need a WT curve as a control for well evaluating the key roles of the residues involved.

We thank the reviewer for highlighting the need to include WT curve controls for better comparisons. We have now added these into Supplementary Fig. 4 and, as can be appreciated, the mutants that lose function are obvious. The EC50 values obtained for each ligand match the published literature (see for example Allen et al., doi: 10.1111/bph.16435 and D'Angelo et al. doi: 10.1016/s0090-6980(96)00091-3).

2. the densities of the agonists should be showed alone for evaluating whether they are fitted properly.

The electrostatic potential maps defining the agonists are now included in the revised version of the manuscript in Supplementary Fig. S2g and Supplementary Fig. S2j for U46619 and I-BOP, respectively. We thank the reviewer for pointing out this omission.

3. although the part "Structural rearrangements that mediate ligand entry" is novel and interesting, the data in this manuscript is not enough to support the hypothesis and a short discussion in this part is needed to clarify that that is just one possibility rather than a final conclusion about the ligand entry. Otherwise, stronger evidence should be provided, because the inactive state can't represent the real apo state (no bound to agonist and G protein), in other word, we don't know whether there is an open gate for ligand entry between TM1 and TM7 in its real apo state conformation similar to that of its inactive state.

We agree wholeheartedly with the reviewer. The text has been modified to clarify that the ligand entry routes referred to in the manuscript are speculative.

4. The last but also the most important, the density map and structure models need to be further improved before it can be published in Nature Communications or other journals. Actually, the densities of Helix 8, ECL and side chains are really poor from the supplementary Fig1., which need more polish, otherwise, they can't support the model properly.

In Supplementary Fig. 1, Helix 8 (H8) cannot be observed, due to the unusual orientation of the helix. In this pose, H8 is oriented with its long axis perpendicular to the viewer and is not visible. The reason for choosing this view was that it shows most structural components in the complex and their qualities. It also reveals scFv16 that is only present in the TP-I-BOP structure. However, the Reviewer is correct to point out that our discussion of H8 and the side chains involved in the binding pocket and beyond, require high resolution structural features that are indeed present in the deposited maps and models. While the published maps and models will be available to the Reader, we realize it is not always easy, especially for the non-expert, to evaluate such features. Accordingly, and based on the Reviewer's well noted point, we have added to the revised manuscript a detailed overview of model in density for all of the structural features discussed. These include individual TM helices and H8 in density (Supplementary Fig. 2f,i), as well as ligand in density (Supplementary Fig. 2g,j) and the ligand binding pocket with side chains in density (Supplementary Fig. 2h,k).

Reviewer #1

- What are the noteworthy results? We can't say the cryo-EM structure as this was published 6 months ago with the same ligand (U46619) and G protein: Li et al., Cell Rep 54 113893 (2024).

The analysis of the mechanism is comprehensive, far more so than Li et al. (2024).

The unbinding mode of the ligand is significant, as it shows exit into the membrane.

We very much appreciate the Reviewer's favourable comments on the analysis of TP's signaling and ligand unbinding mechanisms reported in this study. The revised manuscript includes additional computational data on the mechanism of ligand binding as described below.

- Will the work be of significance to the field and related fields? How does it compare to the established literature? If the work is not original, please provide relevant references.

Unfortunately, the Cryo-EM structure is already published with the same ligand (U46619) and G-protein, so this part may not be significant. (Li et al., Cell Rep 54 113893 (2024), though they did include an extra ligand. In defense of still sending this article to a high-quality journal, they claim 'the question of selectivity and potency are not explained'. It is certainly true that the current article offers a far more comprehensive analysis of the mechanism, which is certainly informative.

We thank the Reviewer for their favourable comments especially those highlighting our work on mechanism.

- Does the work support the conclusions and claims, or is additional evidence needed?

Only simulations of ligand exit have been performed. To justify ligand binding from the membrane, as claimed in the abstract and suggested 5 years ago, the authors should simulate the binding mechanism from the membrane into the receptor, presumably using an enhanced and adaptive sampling method.

We thank the Reviewer for this important suggestion. We have now performed ligand binding simulations using supervised molecular dynamics (suMD), an enhanced and adaptive sampling method, to directly examine membrane-mediated ligand entry into the receptor. These simulations complement our random accelerated molecular dynamics (RAMD) exit studies and provide convincing direct evidence for the proposed binding mechanism. Specifically, we simulated I-BOP agonist binding from the membrane using suMD, starting with twenty independent structures where the ligand was placed in the membrane extracellular leaflet approximately 45 Å from the binding site of the receptor. To avoid potential bias toward the open TM1/TM7 gap observed in antagonist-bound crystal structures, we used the agonist- and G protein-free receptor with a closed TM1/TM7 gap as our starting conformation. Successful suMD trajectories demonstrated that I-BOP can enter the binding site through both TM1/TM7 and TM1/TM2 extracellular interfaces, confirming membrane-mediated ligand entry and highlighting the critical role of TM1 flexibility in the process.

These findings are supported by distance evolution plots showing ligand center-of-mass approach to the binding site across all successful trajectories, along with

structural snapshots depicting key binding events. The suMD results corroborate our RAMD exit pathway analyses and provide the direct binding evidence requested. We have added these findings to the Results section of the main text (the new text is included below) and as a supplementary figure (Supplementary Figure 11).

Additional Text Included in the Revised Manuscript:

“To gain a greater understanding of the mechanism by which the extracellular end of TM1 undergoes the open/closed transition for ligand entry, we performed functional assays on TP mutants alongside random accelerated MD (RAMD)^{45,46,47} and supervised MD (suMD) simulations⁴⁸ (Supplementary Figs. 10-11). RAMD examines ligand-receptor dissociation in response to a random directional force, while suMD guides the ligand toward the binding site. Both enhanced sampling methods explore millisecond-scale interactions on nanosecond timeframes. A sequential strategy was employed whereby RAMD was used to study ligand unbinding (computationally efficient but indirect), followed by suMD to examine direct ligand binding, with unbinding results informing ligand placement outside the receptor.

RAMD studies revealed that bulky antagonists (ramatroban and daltroban) primarily dissociated from the receptor via the TM1/TM7 gap, while flexible agonists (U46619 and I-BOP) showed less specific dissociation patterns using both TM1/TM2 and TM1/TM7 openings with considerable frequency and the extracellular route to a lesser extent (Supplementary Fig. 10). These results are consistent with ligand binding to the receptor that is primarily membrane mediated. They also corroborate results obtained with other GPCRs where TM1-related openings facilitate lipid insertions with possible functional relevance⁴⁹. To avoid bias toward the open TM1/TM7 gap observed in the antagonist-bound crystal structures, we simulated I-BOP agonist binding from the membrane using suMD, where the TM1/TM7 gap is closed. Twenty starting structures were selected from 1- μ s MD simulations of the agonist- and G protein-free receptor, with I-BOP placed in the membrane extracellular leaflet approximately 45 Å from the binding site. Successful suMD trajectories demonstrated that I-BOP entered the binding site primarily through TM1/TM7 (4 out of 6 simulation runs; Supplementary Fig. 11). Notably, in two independent trajectories, a TM1/TM2 opening was observed, where TM1 moved in the opposite direction to that observed with TM7 gate opening, further highlighting TM1 flexibility and the critical role played by TM1 in membrane-mediated ligand binding (Supplementary Fig. 11).”

Additional Supplementary Figure Included in the Revised Manuscript:

Supplementary Figure 11. I-BOP entry pathways through TM1/TM7 and TM1/TM2 gaps in the TP.

a-b suMD simulation snapshots showing I-BOP (carbon atoms, teal spheres) binding to the TP through two distinct membrane entry pathways: (a) via the TM1/TM7 gap and (b) via the TM1/TM2 gap. Transmembrane helices are colored as follows: TM1 (purple), TM2 (blue), and TM7 (orange). Numbers indicate simulation steps progressing from ligand approach to binding site entry. c Distance evolution between centers of mass (CoM) of I-BOP and the TP binding site across 20 independent suMD simulations. The x-axis represents the cumulative number of 150,000-step simulation cycles (both successful and unsuccessful attempts) performed for each system. The y-axis shows the distance between ligand and binding site CoMs. Six simulations successfully reached the 6 Å threshold: four trajectories (red) entered via the TM1/TM7 gap, with two of these achieving close approach (≤ 2 Å), while two trajectories (blue) entered via the TM1/TM2 gap. Simulations that did not reach 6 Å by the time the simulation ended are shown in black.

• Are there any flaws in the data analysis, interpretation and conclusions? Do these prohibit publication or require revision? Nonapparent, apart from the discussion of the W6.48 'toggle switch'. The name 'toggle switch' predates active structures of class A GPCRs which failed to show the conformational change implied by the name, and indeed here, we see a displacement but not a conformational change, so the extensive use of the phrase 'toggle switch' is misleading.

We agree with the Reviewer that use of the term 'toggle switch' can be misleading. Accordingly, the Main and Supplementary texts have been revised. Reference to 'toggle switch' is only made when discussing receptors where the descriptor is appropriate.

• Is the methodology sound? Does the work meet the expected standards in your field?
yes

• Is there enough detail provided in the methods for the work to be reproduced?
yes

Reviewer #2

In this manuscript, Pawel Krawinski and colleagues present two Cryo-EM structures of Gq-coupled TP bound to two analogues of the endogenous ligand, U46619 and I-BOP respectively. The structures suggest Q3017.46 on TM7 may act as the toggle switch in the TP, rather than the highly-conserved residue W2586.48 seen in many class A GPCRs, proposing a unique activation mechanism for the receptor. Therefore, these structures could broaden our understanding for activation of class A GPCRs.

We thank the Reviewer for their favourable comments.

However, I have some concerns about the manuscript.

1. the dose-response curves of BRET assay in Fig 1f and some in supplementary Fig 4 are not entirely convincing with big error bars and each panel of mutant curves also need a WT curve as a control for well evaluating the key roles of the residues involved.

We thank the Reviewer for highlighting the need to include WT curve controls for better comparisons. We have now added these to Supplementary Fig. 4 and, as can be appreciated, the mutants that lose function are obvious. The EC50 values obtained for each ligand match the published literature (see, for example, Allen et al., doi: 10.1111/bph.16435 and D'Angelo et al. doi: 10.1016/s0090-6980(96)00091-3).

2. the densities of the agonists should be showed alone for evaluating whether they are fitted properly.

The electrostatic potential maps defining the agonists are now included in the revised version of the manuscript in Supplementary Fig. 2g and Supplementary Fig. 2j for U46619 and I-BOP, respectively. We thank the Reviewer for pointing out this omission.

3. although the part “Structural rearrangements that mediate ligand entry” is novel and interesting, the data in this manuscript is not enough to support the hypothesis and a short discussion in this part is needed to clarify that that is just one possibility rather than a final conclusion about the ligand entry. Otherwise, stronger evidence should be provided, because the inactive state can't represent the real apo state (no bound to agonist and G protein), in other word, we don't know whether there is an open gate for ligand entry between TM1 and TM7 in its real apo state conformation similar to that of its inactive state.

We thank the Reviewer for this important comment and fully agree that appropriate caveats are essential when discussing ligand entry mechanisms. We have addressed this concern through both methodological rigor and explicit acknowledgment of limitations.

Methodological approach to address apo state uncertainty:

Recognizing that our cryo-EM structures represent agonist-bound states while X-ray structures show antagonist-bound conformations, we designed our suMD simulations to minimize bias toward any particular conformational state. We selected twenty starting structures from 1- μ s MD simulations of the agonist- and G protein-free receptor, allowing the system to sample conformations representative of a ligand-free state. Each structure underwent an additional 5-ns equilibration before ligand placement, further reducing potential artifacts deriving from the initial agonist-bound structure. This approach provides the most unbiased sampling possible given current structural data. However, we acknowledge we cannot and do not claim that it definitively represents the true apo state.

Additional Text Included in the Revised Manuscript:

“Twenty starting structures were selected from 1- μ s MD simulations of the agonist- and G protein-free receptor, with I-BOP placed in the membrane extracellular leaflet approximately 45 Å from the binding site. Successful suMD trajectories demonstrated that I-BOP entered the binding site primarily through TM1/TM7 (4 out of 6 simulation runs; Supplementary Fig. 11). Notably, in two independent trajectories, a TM1/TM2 opening was observed, where TM1 moved in the opposite direction to that observed with TM7 gate opening, further highlighting TM1 flexibility and the critical role played by TM1 in membrane-mediated ligand binding (Supplementary Fig. 11).”

Explicit discussion of limitations:

We have added appropriate caveats to the Discussion section to clarify that our proposed mechanism represents one plausible pathway rather than a definitive conclusion. The discussion now explicitly states:

"While the proposed ligand entry mechanism from MD simulations remains speculative given the uncertainties about the structure/s of the ligand-free receptor, these insights into the structural plasticity of the TP provide new opportunities for designing selective therapeutics."

4. The last but also the most important, the density map and structure models need to be further improved before it can be published in Nature Communications or other journals. Actually, the densities of Helix 8, ECL and side chains are really poor from the supplementary Fig1., which need more polish, otherwise, they can't support the model properly.

In Supplementary Fig. 1, Helix 8 (H8) cannot be observed, due to the unusual orientation of the helix. In this pose, H8 is oriented with its long axis perpendicular to the viewer and is not visible. The reason for choosing this view was that it shows most structural components in the complex and their qualities. It also reveals scFv16 that is only present in the TP-I-BOP structure. However, the Reviewer is correct to point out that our discussion of H8 and the side chains involved in the binding pocket and beyond, require high resolution structural features that are indeed present in the deposited maps and models. While the published maps and models will be available to the Reader, we realize it is not always easy, especially for the non-expert, to evaluate such features. Accordingly, and based on the Reviewer's well noted point, we have added to the revised manuscript a detailed overview of the model in density for all of the structural features discussed. These include individual TM helices and H8 in density (Supplementary Fig. 2f,i), as well as ligand in density (Supplementary Fig. 2g,j) and the ligand binding pocket with side chains in density (Supplementary Fig. 2h,k).

Reviewer #1 (Remarks to the Author):

Review of resubmission

P7, line 165. Demonstrating is too strong because in L78Q, the Q is bigger than L

This is a valid point. If it was Asn instead of Gln, it would be more similar in size to Leu. Therefore, we amended the text: “Accordingly, in our BRET assay, the L78^{2.58}Q mutant showed no activation of G_q signalling suggesting that the hydrophobicity and/or size of the ring-binding subpocket in the TP is important for agonist binding (Fig. 2b and Supplementary Fig. 4).”

P9, 'while SuMD guides the ligand towards the binding site' is a misleading explanation of SuMD that under sells the method (and hence this article) – perhaps the authors should read more references rather than just the original 2014 article.

To address this, we have revised the text and added two additional references. The revised sentence now reads: “RAMD examines ligand-receptor dissociation in response to a random directional force, while SuMD combines short classical MD simulations with a Tabu-like algorithm, which accepts trajectories when the ligand moves closer to the binding site and rejects them otherwise.”

p 11, I am not convinced that W29 undergoes a major conformational transition – as I see it, it is more that TM1 moves.

We agree with the reviewer, and we have revised the text: “W29^{1.37} is a part of the cholesterol binding cavity in the active TP structure (Supplementary Fig. 12e,f). Substitution of the bulky, hydrophobic tryptophan with the smaller, polar cysteine residue may impair receptor association with cholesterol and give rise to membrane localization and signalling defects in individuals harbouring the W29C variant.”

Reviewer #3 (Remarks to the Author):

The authors have incorporated an extensive set of molecular dynamics (MD) simulations in the revised manuscript to substantiate the claims arising from the structural and biochemical

data. This addition significantly strengthens the mechanistic framework of the study. The computational component is clearly described, methodologically sound, and executed with the required scientific rigor. The combination of conventional MD, random accelerated MD (RAMD), and supervised MD (suMD) simulations is appropriate and well-justified. The analyses were internally consistent and aligned with the experimental findings, providing coherent mechanistic insights into ligand binding and activation of the thromboxane receptor. This study is of interest to researchers in the fields of structural biology, computational biophysics, and GPCR pharmacology.

To further enhance transparency and reproducibility, it is recommended that the authors deposit representative MD trajectories, input parameter files, and analysis scripts in a public repository (e.g., Zenodo, Figshare, or GPCRmd) and include accession details in the Data Availability section. This will ensure that the computational work can be independently validated and reused in accordance with current community standards for data sharing in structural and computational biology.

We thank the Reviewer for their favorable comments. MD simulation source data have been deposited to Zenodo and are available at this link: <https://doi.org/10.5281/zenodo.17848900>.